# Shared enhancer gene regulatory networks between wound and oncogenic programs

**Swann Floc'hlay[1,2], Ramya Balaji[3,4], Dimitrije Stanković[5], Valerie M Christiaens[1,2], Carmen Bravo González-Blas[1,2], Seppe De Winter[1,2], Gert J Hulselmans[1,2], Maxime De Waegeneer[1,2], Xiaojiang Quan[1,2], Duygu Koldere[1,2], Mardelle Atkins[6], Georg Halder[7,8], Mirka Uhlirova[5], Anne-Kathrin Classen[3,4]\*, Stein Aerts[1,2]\***

[1]VIB Center for Brain & Disease Research, Leuven, Belgium; [2]Laboratory of Computational Biology, Department of Human Genetics, University of Leuven, Leuven, Belgium; [3]Faculty of Biology, Hilde-Mangold-Haus, University of Freiburg, Freiburg, Germany; [4]CIBSS Centre for Integrative Biological Signalling Studies, University of Freiburg, Freiburg, Germany; [5]Institute for Genetics Cologne Excellence Cluster on Cellular Stress Responses in Aging-Associated Diseases (CECAD), University of Cologne, Cologne, Germany; [6]Department of Biological Sciences, Sam Houston State University, Huntsville, United States; [7]VIB Center for Cancer Biology, Leuven, Belgium; [8]Laboratory of Growth Control and Cancer Research, Department of Oncology, University of Leuven, Leuven, Belgium

**\*For correspondence:**
anne.classen@biologie.uni-freiburg.de (A-KC);
stein.aerts@kuleuven.be (SA)

**Competing interest:** The authors declare that no competing interests exist.

**Abstract** Wound response programs are often activated during neoplastic growth in tumors. In both wound repair and tumor growth, cells respond to acute stress and balance the activation of multiple programs, including apoptosis, proliferation, and cell migration. Central to those responses are the activation of the JNK/MAPK and JAK/STAT signaling pathways. Yet, to what extent these signaling cascades interact at the *cis*-regulatory level and how they orchestrate different regulatory and phenotypic responses is still unclear. Here, we aim to characterize the regulatory states that emerge and cooperate in the wound response, using the *Drosophila melanogaster* wing disc as a model system, and compare these with cancer cell states induced by *ras^V12^scrib^-/-* in the eye disc. We used single-cell multiome profiling to derive enhancer gene regulatory networks (eGRNs) by integrating chromatin accessibility and gene expression signals. We identify a 'proliferative' eGRN, active in the majority of wounded cells and controlled by AP-1 and STAT. In a smaller, but distinct population of wound cells, a 'senescent' eGRN is activated and driven by C/EBP-like transcription factors (Irbp18, Xrp1, Slow border, and Vrille) and Scalloped. These two eGRN signatures are found to be active in tumor cells at both gene expression and chromatin accessibility levels. Our single-cell multiome and eGRNs resource offers an in-depth characterization of the senescence markers, together with a new perspective on the shared gene regulatory programs acting during wound response and oncogenesis.

## Editor's evaluation

This study is an important progression in our understanding of wounding response and its relationship to malignancy. Although this topic has been previously addressed in genetic studies, the use of a systems biology approach here provides compelling support for the dual use of regulatory sequences to achieve context dependence for two linked but non-redundant tasks. Investigators in the fields of gene regulation, developmental biology as well as basic cancer research will find this manuscript to be both important and useful.

## Introduction

The *Drosophila* wing imaginal disc (WID) is a classical model system to study developmental patterning and cell differentiation. This larval primordium is composed of two epithelial cell layers, the peripodial epithelium and the disc proper, along with muscle precursors (*Figure 1a*). Despite its rather complex structure, the wing disc is extensively studied for its regeneration capacities. Damaged wing discs can trigger a set of wound-response mechanisms allowing for disc repair and the formation of normal wings (*Smith-Bolton, 2016*; *Tripathi and Irvine, 2022*). The molecular pathways identified as key drivers for this regenerative process include the regulation of cell apoptosis (JNK, JAK/STAT; *La Fortezza et al., 2016*), cell proliferation (EGFR, Wnt, Wingless, Scalloped; *Blanco et al., 2010*; *Herrera et al., 2013*; *Irvine and Harvey, 2015*; *Smith-Bolton et al., 2009*; *Yu et al., 2015*), re-epithelialization (ERK, Grainy head; *Mace et al., 2005*), and developmental timing (insulin-like peptide 8; *Katsuyama et al., 2015*). Interestingly, the same regulatory pathways may lead to uncontrolled cell apoptosis or neoplastic growth when unrestricted (*La Marca and Richardson, 2020*; *Pérez-Garijo et al., 2013*; *Pinal et al., 2019*). Such tumor-like outcomes are observed in *ras*^V12^*scrib*^-/- transformed cells, where the loss of cell polarity triggers a cellular stress that cells cannot escape via apoptosis (*Atkins et al., 2016*; *Brumby and Richardson, 2003*; *Cosolo et al., 2019*; *Davie et al., 2015*; *La Fortezza et al., 2016*; *Igaki et al., 2006*; *Külshammer et al., 2015*; *Pagliarini and Xu, 2003*; *Pinal et al., 2019*; *Uhlirova and Bohmann, 2006*). In light of these outcomes, it is still unclear how such antagonistic mechanisms (apoptosis and proliferation) interact in the vicinity of a wound to orchestrate tissue repair while controlling for over-proliferation. To study this process at the gene regulatory level, we generated a multi-omic dataset, jointly measuring chromatin accessibility and gene expression changes at single-cell resolution, in wild-type and genetically ablated wing imaginal discs from third-instar larvae (*Figure 1a*).

Leveraging the multidimensionality of our dataset, we constructed enhancer gene regulatory networks (eGRN), centered around transcription factors (TFs) and comprising both gene and enhancer signatures (*Janssens et al., 2022*). We detect two classes of wound populations, respectively expressing markers of proliferation and cellular senescence. The senescent cells are driven by eGRNs belonging to the C/EBP bZip family (Irbp18, Slow border, Vrille; *Blanco et al., 2020*), which we also found to be present in the tumor cells from the *ras*^V12^*scrib*^-/- model.

## Results

### Single-cell multi-omics of the normal and wounded wing imaginal disc

To study the gene regulatory program of a wound response at single-cell resolution, we used a published genetic model that induces a sterile wound (*La Fortezza et al., 2016*; *Smith-Bolton et al., 2009*; see 'Materials and methods'). In this model, the expression of the TNF ligand *eiger* (*egr*) is induced in the wing pouch, where *rotund* (*rn*) is expressed, at specific times through a temperature shift regime. The expression of *egr* causes a wound by inducing extensive local cell death (*Figure 1a*). Wing discs were dissected 24 hr after the induction of *rn*-Gal4, the time point when most *rn*-expressing cells have undergone apoptosis, and markers of wound response are observed (*Herrera et al., 2013*; *Pérez-Garijo et al., 2013*). After disc dissociation, we performed multiome experiments using 10X Genomics (combined scRNA & scATAC from the same cell), as well as on wild-type control discs. This yielded 17,402 high-quality cells, with a median of 1124 detected genes per cell, and a median of 3060 unique ATAC fragments per cell (*Figure 1—figure supplement 1*).

To increase the power to robustly detect cell types and simultaneously validate our single-cell data, we integrated our scRNA dataset with several previously published scRNA-seq datasets of wild-type wing imaginal discs, generating an integrated atlas. This wing atlas is available through SCope, along with clustering and marker gene information (*Figure 1b*, *Figure 1—figure supplement 2a*, https://scope.aertslab.org/#/WingAtlas/*/welcome). The integrated atlas contains 70,230 cells from 10X Genomics, across 10 replicates (four from this work, four from *Everetts et al., 2021*, one from *Bageritz et al., 2019*, and one from *Deng et al., 2019*).

To annotate the cell types of the wild-type disc, we compared markers from literature with significantly upregulated gene sets obtained from each cluster (see 'Materials and methods'), resulting in the identification of the previously known wing disc cell types and patterning domains (*Figure 1c*, *Supplementary file 1*). We also confirmed our annotation by comparison with the annotations from the integrated public datasets (*Figure 1—figure supplement 2b*). The annotated clusters form a

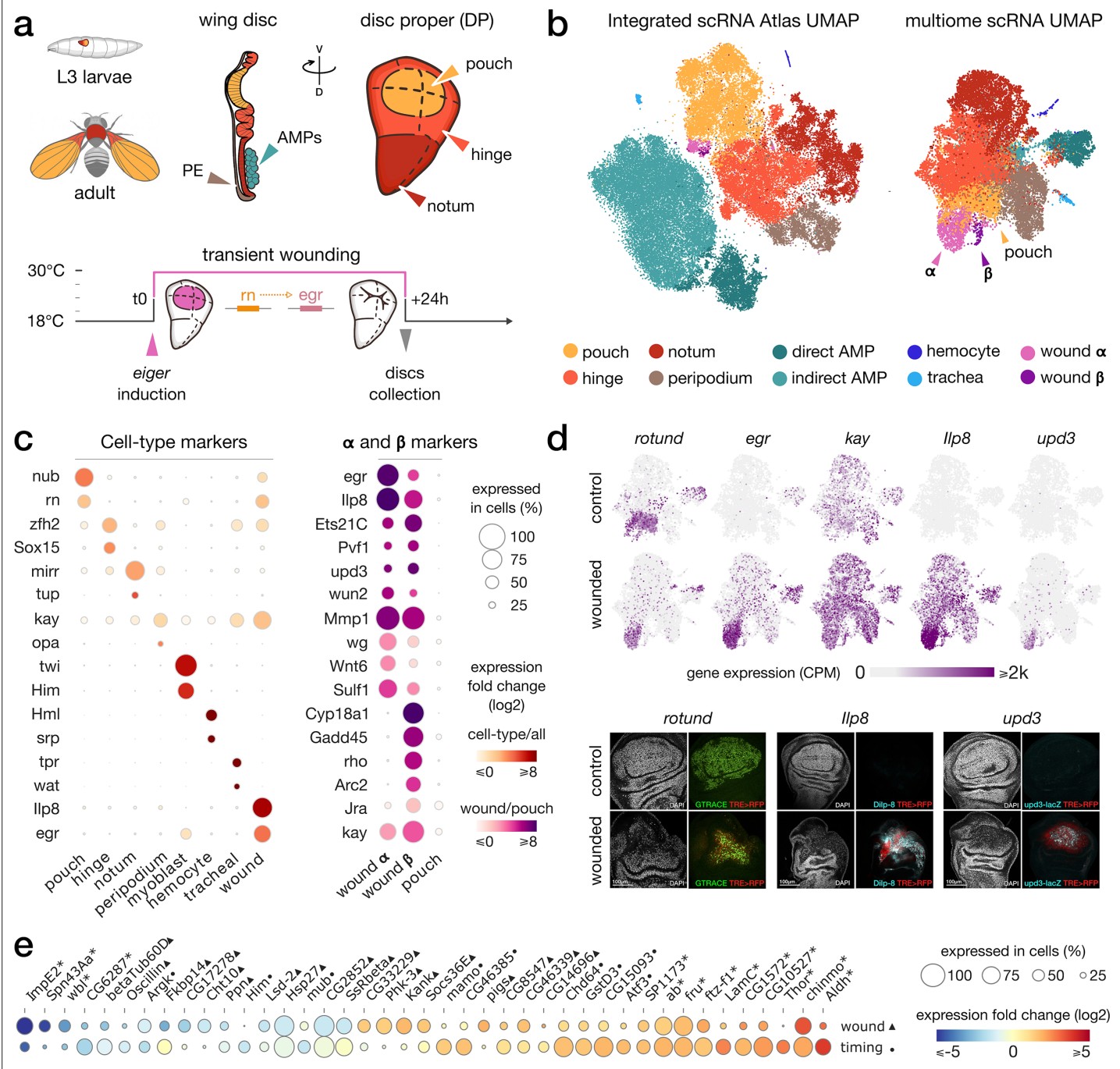

**Figure 1.** Gene expression patterns in wild-type and wounded wing discs. (**a**) Top: schematic of the wing imaginal disc subdomains from a *Drosophila* third-instar larvae. The disc proper (DP) is composed of three domains and is shoehorned between the peripodial epithelium (PE) and the adult muscle precursors (AMP, or myoblasts). bottom: Design of the wound experiment, *eiger* expression is induced in the pouch for 24 hr, resulting in a localized apoptosis. (**b**) Representation of the wing disc scRNA Atlas tSNE (left) and the scRNA multiome data UMAP (right), where wound populations α and β can be detected. (**c**) Gene expression of marker genes across cell types (left) and wound subclusters in contrast to pouch (right) clusters. (**d**) Gene expression pattern of wound markers in multiome UMAP (top) and immunostaining (bottom). *Ilp8*, *rn* and *upd3* are found to colocalize with JNK reporter (TRE-RFP), present at the wound site. (**e**) Relative expression (log2FC) of markers of wound response (first row, wounded/wild-type, ▲) and/or developmental timing (second row, 96 hr/120 hr after egg laying, ●). Shared markers are marked by an asterisk (*).

The online version of this article includes the following figure supplement(s) for figure 1:

**Figure supplement 1.** Single-cell quality metrics of multiome experiments.

**Figure supplement 2.** integration of scRNA and scATAC samples.

continuum of epithelial cells, from the pouch over the hinge to the notum (globally marked by the epithelial marker *grainyhead*); and separate clusters of myoblasts (marked by myogenic genes such as *twist, holes in muscle* and *zn finger homeodomain 1*), hemocytes, and tracheal cells (*Figure 1b*, *Figure 1—figure supplement 2c*).

One cluster was noticeably enriched for cells from the wounded disc samples, with markers linked to stress response pathways (e.g. *kayak, insulin-like peptide 8, unpaired3*, p-adj <10e⁻³, log2FC >1.7; *Figure 1c*, *Figure 1—figure supplement 2d*) and localized at the wound site in the pouch domain (*Figure 1d*), suggesting that this cluster represents a wound-response cell state. This cluster can furthermore be subdivided at higher resolution into two distinct cell populations (population **α** of 1211 cells and population **β** of 94 cells, *Figure 1b and c*).

Next, we analyzed the scATAC-seq part of our multiome dataset separately using cisTopic (*Bravo González-Blas et al., 2019*; *Figure 1—figure supplement 2e*). Since multiome data delivers same-cell measures for RNA and ATAC, we could label the scATAC-seq based on the previously derived scRNA-seq annotations. The detected ATAC clusters, based solely on chromatin accessibility, also identify the hinge, pouch, notum, myoblast, peripodial epithelium, and wound clusters (*Figure 1—figure supplement 2f*). Thus, both chromatin accessibility and gene expression independently identify normal cell types and a separate wound cell state.

## The wounded disc is delayed in its developmental timing

The proper regeneration of a damaged wing disc is tied to the introduction of a developmental delay via a reduced ecdysone signaling. This delay provides the necessary time for tissue repair before pupation (*Jaszczak and Halme, 2016*; *Katsuyama et al., 2015*; *Sanchez et al., 2019*). To assess whether we can detect this regulatory response in our integrated dataset, we combined cells from multiple conditions (wounded/wild-type) but also normal discs dissected at different developmental time points (96 and 120 hr after egg laying, from *Everetts et al., 2021*). By comparing the up and downregulated genes with respect to the developmental time, we found that marker genes of late time points (e.g. *ecdysone-inducible gene E2*) are globally downregulated in the wounded disc samples dissected at the same time point. We furthermore found a significant overlap of downregulated markers for both wound response and developmental timing (21%, p-adj <10e⁻³, Fisher's exact test, *Figure 1e*). We confirmed this result when restricting the analysis outside of the wound site, in the notum domain (34% shared downregulated genes, p-adj <10e⁻³, Fisher's exact test, *Figure 1—figure supplement 2g*). This resemblance of wounded disc cells with those from earlier stage wild-type larvae confirms that wounding triggers a global reaction across the whole disc which delays development to give more time for the wound to repair completely before metamorphosis. This delay is likely driven by *insulin-like peptide 8* (*ilp8*), a critical long-range regulator of ecdysone signaling, highly expressed in the wound cluster (*Figure 1c and d*).

We additionally found genes significantly upregulated in the entire wing disc following wounding, with no strong change between developmental time points (p-adj <10e⁻³, log2FC_wound > 1.7, log2FC-timing < 0.5, *Figure 1e*). Among these wound markers, *suppressor of cytokine signaling at 36E* (*socs36E*) is a known target of the JAK/STAT and EGFR pathways (*Berez et al., 2020*; *Kang et al., 2018*). We also noticed an upregulation of *pickled eggs* (*pigs*), a potential Notch regulator (*Pines et al., 2010*), in the hinge and peripodial cells of wounded discs.

## Multi-ome gene regulatory network reconstruction

An important advantage of single-cell multiomics data is the power to detect regulatory interactions by synchronous changes in expression and/or accessibility across cells (*Fiers et al., 2018*). Here, we set out to infer enhancer-GRNs (eGRN) following a similar strategy as we recently applied to the fly brain (*Janssens et al., 2022*; *Figure 2a*, see 'Materials and methods'). First, we defined differentially accessible regions (DARs) for each cell type, including the wound cluster, using cisTopic (*Bravo González-Blas et al., 2019*). Next, these DARs were tested for enrichment in TF binding motifs using cisTarget (*Herrmann et al., 2012*), which resulted in a list of TF-to-regions edges (i.e. regulatory links) with a significant motif hit. Next we determined co-variability between gene expression and accessibility of neighboring enhancer regions to generate a list of region-to-gene edges. We then completed the edges loop by using the TF-to-gene adjacency scores from *pySCENIC* (*Van de Sande et al.,*

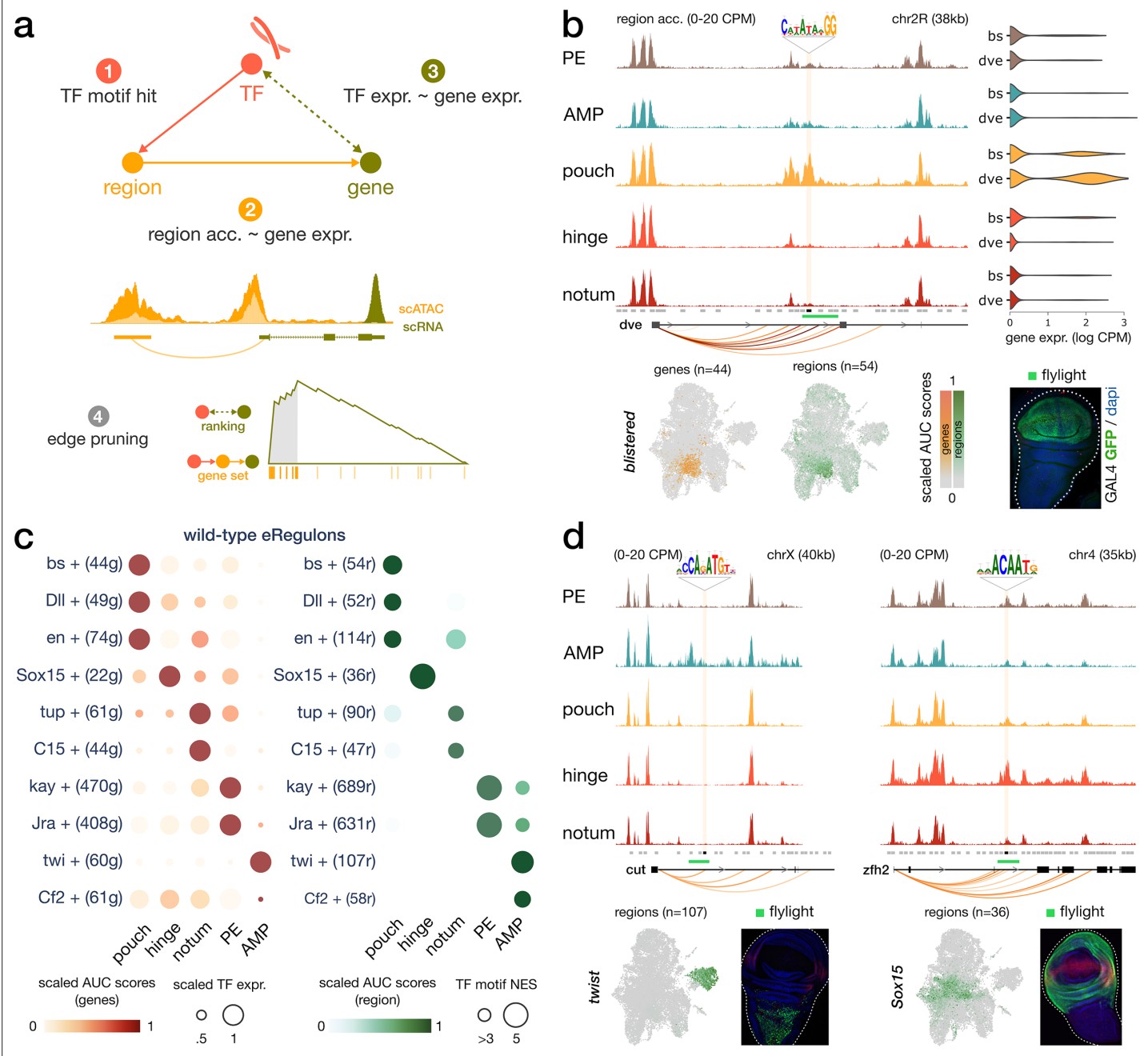

**Figure 2.** Construction of enhancer-mediated gene regulatory networks (eGRNs). (**a**) eGRN construction is based on TF motif enrichment (1), co-variability of gene expression and chromatin accessibility (2–3), and followed by functional edge selection (4). (**b**) Aggregated wild-type chromatin accessibility signals around a target region (black rectangle, gold shaded box) of the *blistered* eGRN. The target region comprises a *blistered* motif hit, it is significantly correlated with expression of *defective proventriculus* (*dve*, orange arcs), and it overlaps a flylight reporter (green rectangle) expressed in the pouch domain. The activities of the *blistered* TF and its associated eGRN signatures are all similarly localized in the pouch (UMAP and violin plots). (**c**) Dotplot of the average gene-based (left) and region-based (right) activity of selected eGRNs with highest cell type specificity in wild-type conditions. (**d**) Aggregated chromatin accessibility signals similar to panel (**b**) for the target regions of two other eGRNs, active in the myoblast (*twist*, left) and the hinge domains (*Sox15*, right).

The online version of this article includes the following figure supplement(s) for figure 2:

**Figure supplement 1.** Enhancer gene regulatory network (eGRN) specificity across wild-type wing disc domains.

*2020*). Lastly, we filtered for high-quality TF-to-region-to-gene interactions by keeping the leading edges of a gene set enrichment analysis (GSEA), taking the TF-to-gene scores as ranking values.

This eGRN inference approach has the advantage of including distal enhancers, found up to 50 kb away from a putative target gene. This procedure resulted in 147 high-quality eGRNs (85 activating, 62 repressing), spanning 98 TFs, with on average 54 target genes and 58 target regions (*Supplementary file 2*). An example of an inferred eGRN is the TF *blistered* (*bs*), which is predicted to target the repressor *defective proventriculus* (*dve*) via multiple intronic enhancers (*Figure 2b*). Using the target genes and target regions sets as proxies, we scored the activity of a TF and its subsequent domain specificity via its entire eGRN by *AUCell* (*Figure 2c*, see 'Materials and methods'). In the case of *bs*, we found the eGRN to be specifically active in the pouch, both from the gene expression and the region accessibility perspectives (*Figure 2b and c*).

We can further divide our final list of eGRNs into repressor and activator categories, based on linear correlation between TF expression and accessibility. For example, we identified *mirror* (*mirr*) as a repressor TF in control wing discs because its expression negatively correlated with accessibility of both its target genes and regions (*Figure 2—figure supplement 1a*). The repressive action of Iroquois TFs like *mirr* has already been demonstrated in *Drosophila* (*Andreu et al., 2012*; *Bilioni et al., 2005*).

We used the TF activity scores derived from gene expression and chromatin accessibility to compute the regulon specificity score (RSS) of each eGRN across the disc domains (*Suo et al., 2018*). A high RSS score indicates an enrichment for the TF signature (gene or region targets) among the top markers of a given domain. One of the strongest regulatory programs was observed in the hinge with the Sox box protein 15 (Sox15) (*Dichtel-Danjoy et al., 2009*) eGRN (22 genes, 36 regions, *Figure 2c and d*) where the top predicted target genes included *zn finger homeodomain 2*, *frizzled*, *dachsous*, and *homothorax*. Several other well-known wing development TFs were identified, including *tailup* and *odd-paired* in the notum, *ultrabithorax* and *C15* in the peripodial epithelium and *twist* in the myoblasts (*Figure 2c and d*, *Figure 2—figure supplement 1a–c*). Another interesting example is *nubbin* (*nub*), a POU/homeodomain transcription factor targeted by *vestigial* (*Rodríguez D del Á et al., 2002*), which is found specifically expressed in the pouch domain (*Figure 2—figure supplement 1a*). An extended list of TFs and their target genes can be queried via SCope (https://scope.aertslab.org/#/WingAtlas/*/welcome), and is provided as a *Supplementary file 2*.

## The wound site shows a strong JNK and JAK/STAT eGRN activity

Both α and β wound clusters are unique to the wounded disc and display markers of wound response at both gene expression and chromatin accessibility levels. These two clusters are associated to 3980 enhancers (DARs, see 'Materials and methods,' *Supplementary file 3*), among which 24% have been previously described as damage-responsive regulatory elements (*Harris et al., 2020*; *Vizcaya-Molina et al., 2018*), including the *BRV118* locus (*Gracia-Latorre et al., 2022*; *Figure 3—figure supplement 1*). Their signatures at the gene expression level also share enrichment for GO terms related to wound response and paracrine signaling (p-adj <10e$^{-3}$).

The two wound clusters express high levels of stress-response genes, including *ilp8* (*Katsuyama et al., 2015*), *matrix metalloproteinase 1* (*Mmp1*; *Harris et al., 2020*), *moladietz* (*mol*; *Khan et al., 2017*), *PDGF- and VEGF-related factor 1* (*pvf1*; *Wu et al., 2009*), and *jun-related antigen/kayak* (*jra/kay*, homologs of JUN and FOS, forming the AP-1 complex, involved in the JNK cascade); (*Cosolo et al., 2019*; *Figure 1c and d*), (p-adj <10e$^{-3}$, log2FC > 1). The activity of the JNK pathway is further confirmed by a high and specific activity of the Jra/Kay eGRNs in the wound populations, at both gene expression and chromatin accessibility levels (*Figure 3a*). We also confirm the strong involvement of JAK/STAT as the eGRN of Signal-transducer and activator of transcription protein at 92E (Stat92E) is specifically active in the wound response clusters (*Figure 3a*), in agreement with the enrichment for the Stat92E binding motif in the wound-specific accessible regions (NES = 3.6) and the co-expression of the *unpaired* ligands (*upd1/2/3*, *Figure 1c*).

Other transcription factors with induced eGRN activity in both wound clusters include the Cyclic-AMP response element binding protein A (CrebA, CREB3L2 ortholog, involved in resistance to infection), Cap'n'collar (Cnc, Nrf1/2 ortholog, involved in oxidative stress response), and Cryptocephal (Crc, ATF4 ortholog, involved in unfolded protein response) (*Figure 3a*), three basic-leucine zipper (bZip) TFs known to be involved in stress response (*Brock et al., 2017*; *Brown et al., 2021*; *Ragheb et al., 2017*; *Sorge et al., 2020*; *Sykiotis and Bohmann, 2008*; *Troha et al., 2018*). Interestingly, we

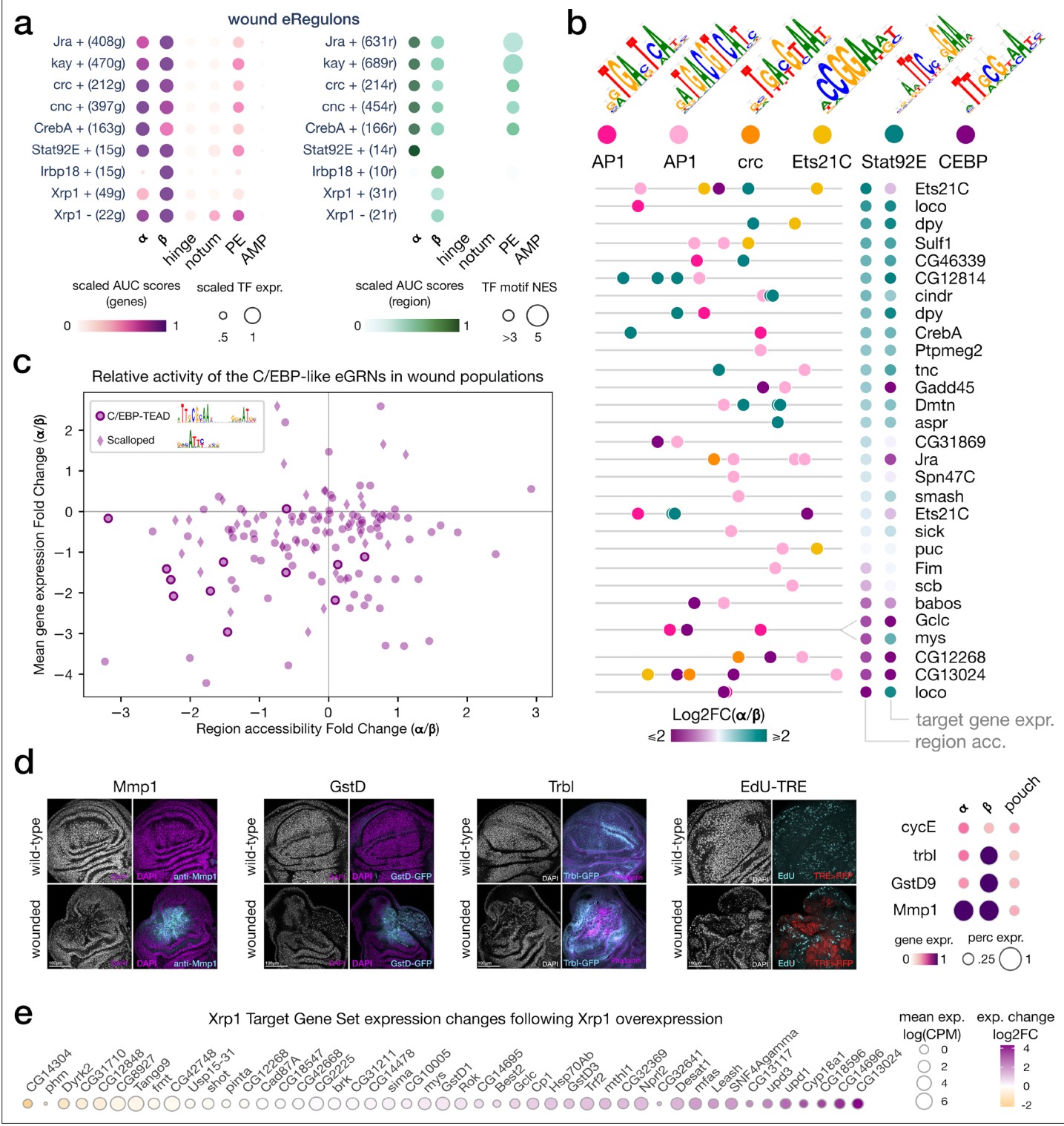

**Figure 3.** bZIP TF activity in senescent and proliferative wound populations. (**a**) Dotplot of the average gene-based (left) and region-based (right) activity of selected enhancer gene regulatory networks (eGRNs) with highest α and/or β specificity in wounded conditions. (**b**) Feature maps of six types of TF motifs on regions specifically accessible in the wound populations α and/or β. AP1 bindings are homogeneously present, while Stat92E and C/EBP motifs are specifically enriched in wound population α and β, respectively. One region targeted by both AP1 and C/EBP GRNs regulates two genes with antagonistic α/β expression patterns (mys and Gclc). (**c**) Scatterplot of the relative changes in target gene expression and chromatin accessibility for all enhancers targeted by the C/EBP eGRNS (vri, Irbp18, Xrp1, slbo). We note the presence of the CEBP-TEAD dimer motif in regions strongly upregulated in β. (**d**) Wing disc immunostaining (left) and normalized average expression of three wound marker genes and a proliferation marker (CycE, right). Both

*Figure 3 continued on next page*

Figure 3 continued

α and β marker genes are expressed and localized at the wound site. (**e**) Expression change of Xrp1 eGRN target genes following Xrp1 overexpression, we note the strong upregulation of the Unpaired ligands.

The online version of this article includes the following figure supplement(s) for figure 3:

**Figure supplement 1.** Damage-responsive element BRV118.

**Figure supplement 2.** Contrast between wound and regenerating populations.

**Figure supplement 3.** Enrichment of the C/EBP-TEAD motif in senescent cells.

**Figure supplement 4.** Gene set enrichment analysis of the Xrp1 overexpression data.

**Figure supplement 5.** Single-channel Immunostainings.

note an additional upregulation of both Stat92E and AP-1 eGRN activity in the peripodial epithelium (PE) of the wounded disc (*Figure 3a*). Although the PE was not directly targeted by the genetic ablation, this domain is in close proximity with the pouch territory in wild-type wing discs (*Figure 1a*). We therefore hypothesized that stress signaling is capable of local diffusion across epithelial layers. We further identified a JAK/STAT repressor *apontic* (*apt*), to be specifically expressed in the peripodial epithelium in wounded conditions (p-adj < $10e^{-3}$, log2FC > 1). This finding supports a protective role of Apontic to block a response to Stat92E proliferative signaling in the vicinity of wounded tissues (*Harris et al., 2020*).

Taken together, our results highlight JNK and JAK/STAT as the most prominent markers of the global wound response program, shared by the two cell populations **α** and **β**. AP1 is the largest inferred wound eGRN (470 target genes, 689 target regions) and its binding motifs (CRE : TRE variants, *Fonseca et al., 2019*) are strongly enriched in DARs from both wound populations **α** an **β** (*Figure 3b*).

## Proliferative and senescence signals separate the α and β populations

In the wound samples, we expect the *rn*-expressing pouch cells to be almost entirely lost upon apoptosis from the *rn*-Gal4 induction. However, we find persistent *rn*-positive cells in the wound cluster α (p-adj < $10e^{-3}$, log2FC > 2) that are excluded from cluster β (p-adj$_{α/β}$ < $10e^{-3}$, log2FC$_{α/β}$ > 2). These cells are distinct from the normal *rn*-positive pouch cells though as they also show stress response markers. We further observe an upregulation of markers of tissue patterning and proliferation in cluster α relative to cluster β, like *wingless* (*wg*), *Wnt oncogene analog 6* (*Wnt6*) (*Figure 1c*, p-adj$_{α/β}$ < $10e^{-3}$, log2FC$_{α/β}$ > 1) and *CyclinE* (*CycE*) (*Figure 3d*, log2FC$_{α/β}$ > 0.8). Consistent with the higher enhancer activity of the Stat92E eGRN in wound α compared to β (*Figure 3a and b*), we find a significant overlap between the α marker genes and a study that identified JAK/STAT as coordinating cell proliferation during wing disc regeneration (GSEA, NES = 1.292) (*Katsuyama et al., 2015*). We also find evidence for an upregulation of the pro-regenerative marker *Ets at 21C* (*Ets21C*) (*Worley et al., 2022*), with a combined enrichment of its gene expression (p-adj < $10e^{-3}$, log2FC > 8) and an enrichment of the Ets21C binding motif in cluster α (*cistarget*, NES = 3.35). In line with *Ets21C* upregulation, we integrated our data with the study from *Worley et al., 2022* that focused on disc regeneration and found strong similarities between their reported regenerative blastemas and our wound population **α** (*Figure 3—figure supplement 2*). These results demonstrate that pro-proliferative and pro-regenerative characteristics are specific to the α population in the wound.

In contrast, cells from cluster β do not show clear proliferative nor regeneration markers. Genes found upregulated in β compared to α include genes associated with innate immunity (*Dorsal-related immunity factor*), glutathione metabolism (*Glutamate-cysteine ligase catalytic subunit, Glutathione S transferase D9*)(p-adj$_{β/α}$<$10e^{-3}$, log2FC$_{β/α}$ > 2), cell migration signaling (*e.g rho1, slow border* and *pvf1*, p-adj <$10e^{-3}$), response to irradiation (e.g. *inverted repeat-binding protein*, p-adj <$10e^{-3}$) and negative regulation of cell cycle (e.g. *growth arrest and DNA damage-inducible 45, tribbles*). Among these markers, *tribbles* (*trbl*) and *slow border cells* (*slbo*) encode a known repressor and a target of the JAK/STAT pathway in border cell migration, respectively (*Berez et al., 2020*; *Dobens et al., 2021*; *Harris et al., 2020*; *Rørth et al., 2000*; *Starz-Gaiano et al., 2008*). The cells expressing these markers are located at the center of the wounded pouch, despite their lack of pouch-specification markers (*Figure 3d*). These results suggest that cluster β contains cells derived from the wild-type, *rn*-expressing domain that have de-differentiated and hence no longer express wing disc markers.

In addition to their loss of wing fate, the cells from cluster β show interesting similarities with cellular senescence. Indeed, a recent study in wounded wing imaginal discs associate the emergence of senescent-like cells with the presence of markers of stress-response (*jra*, *kay*), DNA-damage response (*gadd45*), paracrine signaling (*pvf1*), and cell cycle stalling (*trbl*), as observed in our population β (*Cosolo et al., 2019*; *Jaiswal et al., 2022*). Precisely, one marker of our cluster β population, *tribbles* (*trbl*), was found to be partly responsible for cell cycle stalling in wounded cells. Nevertheless, it still remains unclear whether these wounded cells will later die or whether they will revert to the pouch disc fate after wounding (*Jaiswal et al., 2022*). Hence, we refer here to cellular senescence and associated secretory phenotype in terms of gene expression programs rather than evidence of a terminal cell cycle arrest. The *slbo* expression in population β further corroborates the proximity with senescence-associated secretory phenotype (SASP) as *slbo* is homologous to the human C/EBPB homodimer, a major mediator of oncogene-induced senescence (*Kuilman et al., 2008*; *Lee et al., 2010a*; *Lee et al., 2010b*; Reactome ID R-DME-2559582). Together with the lack of pouch markers, our results suggest that wound-response population β corresponds to cells that have lost their initial pouch fate, arrested cycling and express classical senescence markers.

## The senescent population is characterized by C/EBP eGRN activity

By examining eGRN activities, we find several TFs with a strong specificity score for the senescent population (β), namely the heterodimer partners *inverted repeat binding protein 18 kDa* (*irbp18*) and *xrp1*, *slbo*, *vrille* (*vri*), and *sox box protein 14* (*sox14*) (top 6% RSS, *Figure 3a*). Among them, the two bZip TFs *irbp18* and *vri* are significantly upregulated in the senescent population compared to the proliferative population (α) (p-adj $<2 * 10e^{-3}$, $log2FC_{\beta/\alpha}>2.4$) and share similarities in their TF binding motifs with the mammalian, senescence-associated C/EBP proteins (*Figure 3b*; *Blanco et al., 2020*). The Irbp18 eGRN (extended category, see method) is composed of 149 predicted target regions (80 wound-specific and 11 senescent-specific regions) and 116 target genes (52 markers of senescent population, p-adj $<5 * 10e^{-3}$, $log2FC > 1.5$) including *slbo, vri, ets at 98B* (*ets98B*), *socs36E, head involution defective* (*hid*), *growth arrest and DNA damage-inducible 45* (*gadd45*) and *p53*. The presence of these TFs in the senescent cluster (β) is further corroborated by the significant enrichment for C/EBPB-like binding motifs in senescent-specific peaks (NES = 5.25, *Figure 3b and c*) generated using MACS2 *bdgdiff* (*Zhang et al., 2008a*).

Our eGRN approach has an important limitation, namely that TFs can be identified only if their expression co-varies with chromatin accessibility and target gene expression. A key TF that remains undetected in the eGRN approach is Scalloped (Sd). This homolog of mammalian TEAD factors is the effector TF of the Hippo signaling pathway (*Zhang et al., 2008b*). Despite the absence of *Sd* mRNA upregulation, nuclear Sd protein may increase in the senescent population since we find a significant enrichment for the Sd motif in the senescent-specific accessible regions (NES = 4.89). In fact, the top enriched motif in these senescent-specific regions is a C/EBP-TEAD dimer motif (NES = 7.48, *Figure 3c*), notably found in the enhancer of CG13024, a strong marker of both senescent and *ras^V12 scrib^-/-* cells (*Figure 3—figure supplement 3*, see next section).

Our results suggest that the eGRN activity of C/EBP orthologs, such as the Irbp18-Xrp1 heterodimer, is a strong marker of cellular senescence. To confirm the activity of C/EBP eGRNs in our wound populations, we used bulk RNA sequencing to compare gene expression changes induced by a tem

poral overexpression of the short *Xrp1* isoform (*Xrp1-S*) in the developing wing imaginal discs using *rn^ts* >driver (*rn^ts >Xrp1S*) in comparison to control discs (*rn^ts*>crossed to w^1118; see 'Materials and methods'). The 200 significantly upregulated genes (DEseq2, logFC > 1.5, p-adj $< 5.10e^{-3}$) in response to Xrp1 overexpression were strongly enriched for markers of cellular senescence (Reactome Pathway Database, *Gillespie et al., 2022*) and α and β wound populations. As an additional validation of our eGRN construction, we also find the inferred target gene sets of C/EBP eGRNs significantly enriched among the upregulated genes (*Figure 3e*, *Figure 3—figure supplement 4*).

Given that these eGRNs target multiple markers of cell migration, cell cycle stalling, and DNA damage response, we hypothesize that C/EBP signaling may be responsible for establishing a state of cellular senescence in the wound vicinity. By transiently escaping cell death, such cells play an essential role in stress response induction and may serve as a 'flagship' population to lead the regenerative cells toward wound closure (*Cosolo et al., 2019*; *Kozyrska et al., 2022*). The list of all significant marker genes for both wound populations is available in the *Supplementary file 4*.

## Shared regulatory programs between wound response and cancer

Multiple pathways associated with senescence, such as JNK and JAK/STAT, have been found to be associated with tumorigenesis when their activity becomes unrestricted (*Atkins et al., 2016*; *Cosolo et al., 2019*; *Davie et al., 2015*; *La Fortezza et al., 2016*; *Külshammer et al., 2015*; *Pinal et al., 2019*; *Uhlirova and Bohmann, 2006*). One oncogenic system widely used is the *ras^V12^scrib^-/-* tumor model (*Brumby and Richardson, 2003*), where the *scribble^-/-^* (*scrib^-/-^*) mutation causes epithelial cells to lose polarity, while overexpression of *ras85D^V12^* (*ras^V12^*) prevents them from being outcompeted (*Figure 4a*). The constant stress caused by tissue disorganization, combined with the reduction of apoptosis, leads to an overgrown population of cells in the eye-antennal disc (*Atkins et al., 2016*; *Davie et al., 2015*; *Külshammer and Uhlirova, 2013*), described as *aberrant senescence* (*Cosolo et al., 2019*; *Rhinn et al., 2019*). In order to compare the senescent state (β) observed in our transient wound model with tumor overgrowth, we performed scRNA-seq of *ras^V12^scrib^-/-* eye-antennal discs (6717 cells). We combined this dataset with publicly available scRNA-seq data on wild-type eye-antennal discs (*Bravo González-Blas et al., 2020*) (4830 cells), scRNA-seq data on 14-day *scrib^-/-^* imaginal wing discs (*Deng et al., 2019*; *Ji et al., 2019*) (7554 cells), and bulk ATAC-seq data from *ras^V12^scrib^-/-* tumors (*Davie et al., 2015*). Altogether, these datasets allowed us to compare wound response programs across different model systems (eye and wing discs), regulatory layers (RNA and ATAC), and conditions (following wound induction and during persistent oncogenic induction). In this context, we believe the cell-of-origin has minor to no effect, given that previous RNA-seq experiments performed in *ras^V12^scrib^-/-* eye, antennal, wing, and leg discs showed strong overlap of ectopically expressed genes (*Atkins et al., 2016*). This was also confirmed by recent in vivo experiments showing similar marker gene expression in the *ras^V12^scrib^-/-* models of both eye-antennal and wing imaginal discs (*Jaiswal et al., 2022*).

We first characterized regions specifically accessible in the *ras^V12^scrib^-/-* model compared to wild-type eye discs (*Figure 4b*). In agreement with *Davie et al., 2015* this signature is enriched for the binding motifs of Stat92E (NES = 4.19) and AP-1 (NES = 7.17). Looking at pseudo-bulk ATAC aggregate wing disc data, we noted a significant enrichment for the *ras^V12^scrib^-/-* signature regions for both proliferative (**α**) and senescent (**β**) wound populations in contrast to wild-type pouch cells (p-adj < 10e⁻³, one-tailed *t*-test, *Figure 4b*). In line with the observed AP-1 and Stat92E activity in wounded peripodial epithelium, we also find the *ras^V12^scrib^-/-* signature enriched in this domain (*Figure 4—figure supplement 1a*). These similar chromatin accessibility profiles between wound and tumor populations suggest at least part of the *ras^V12^scrib^-/-* cells also activate the senescence and proliferative programs.

We further investigated the similarities with oncogenic induction at the gene expression level by integrating all single-cell RNA datasets into a combined, batch-corrected, analysis (44,759 cells, see 'Materials and methods'). For each oncogenic model (*ras^V12^scrib^-/-* and *scrib^-/-^*), we detect a tumor-specific cluster, mostly composed of cells originating from samples with induced tumorigenesis (*Figure 4c*). This analysis reveals a co-clustering of the senescent wound cells (β) with *ras^V12^scrib^-/-* and *scrib^-/-^* cells (*Figure 4c*). We confirm the similarity in gene expression patterns between oncogenic models and both wound populations by computing the activity of the proliferative and senescent signatures (**α** and **β** relative to control pouch) in wild-type and oncogenic conditions (*Figure 4d*). We find both signatures to be upregulated, with the strongest increase in gene expression between wild-type and the two oncogenic models for the senescent signature (p-adj < 10e⁻³, one-tailed *t*-test, *Figure 4d*). We find a similar result for signatures contrasting the proliferative and senescent populations against each other (**α** vs. **β**), further supporting the closest resemblance between the chronic oncogenic response and senescent population (*Figure 4e*, *Figure 4—figure supplement 1b*).

Unlike the wound response, we do not observe two distinct proliferative-like and senescent-like cell populations in tumors. Instead, the tumor population seems to activate both pro-proliferative and pro-senescence programs in an aberrant combination. To further explore cellular heterogeneity in *ras^V12^scrib^-/-* tumors, we performed a sub-clustering analysis and detected four tumor subpopulations, delimited by the expression gradient of several markers of wound β and C/EBP targets genes such as Gadd45 and CG13024 (*Figure 4—figure supplement 2*, *Supplementary file 5*). We also found cellular senescence markers, including 6 of the 10 leading edge genes detected in the Xrp1 overexpression GSEA. Despite higher noise levels in the tumor clustering analysis, the detected cellular heterogeneity in *ras^V12^scrib^-/-* tumors would corroborate the results from a recent study by *Jaiswal et al., 2022*, which depicts the emergence of cellular heterogeneity in early-stage *ras^V12^scrib^-/-* tumors

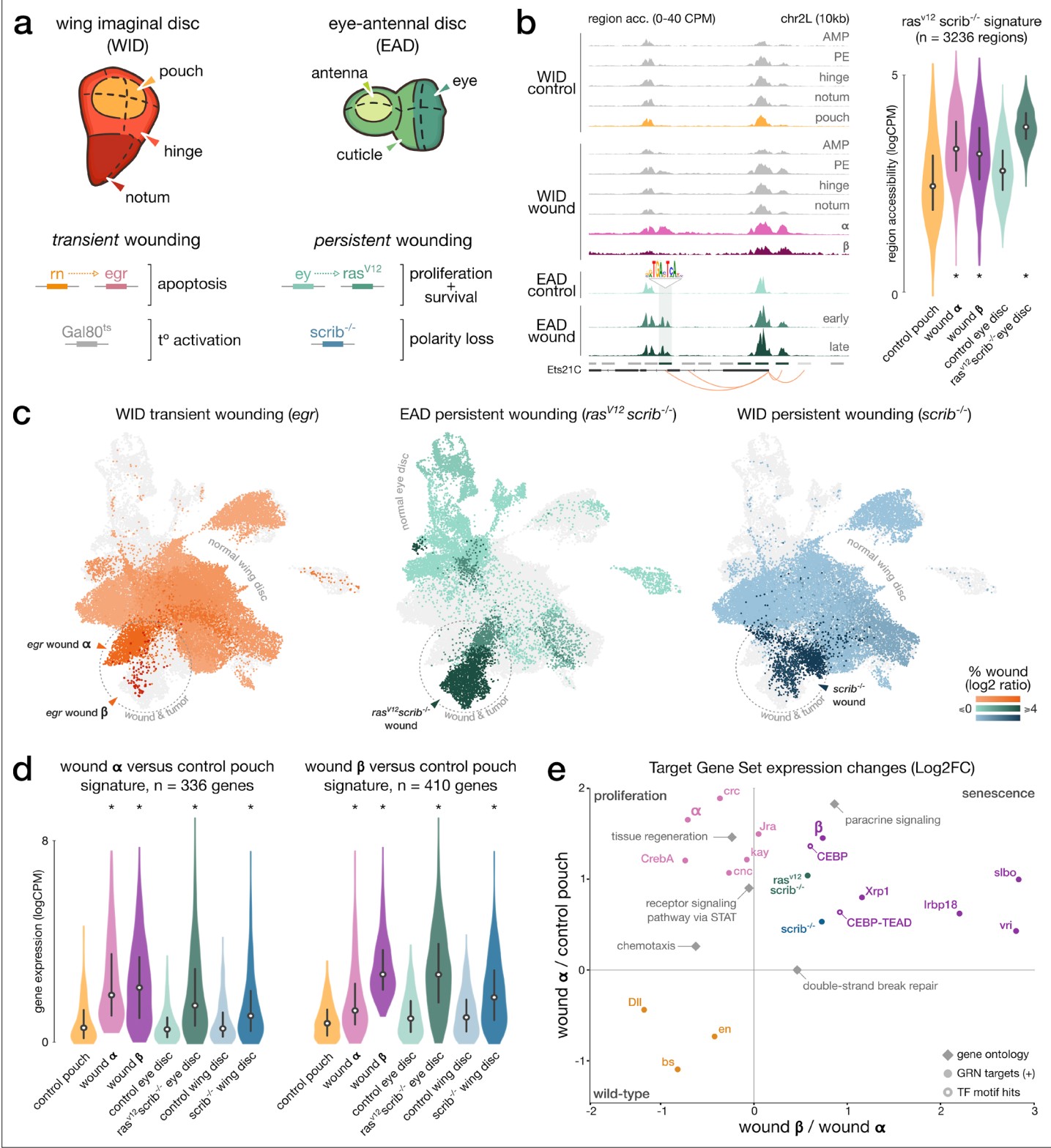

**Figure 4.** Shared signatures between transient and persistent wound induction. (**a**) Top: schematic of the imaginal wing and eye-antennal discs. bottom. Genetic constructs used to trigger localized, transient, or persistent wounding. (**b**) Left: example of a wound-responsive locus: aggregated (WID) and bulk (EAD) chromatin accessibility signals in wild-type and wound conditions at Ets21C loci, including an AP-1 target region (TF motif). right. Distribution of ATAC measures in *ras^V12^scrib^-/-^* region signature; asterisks denote significant enrichment compared to wild-type. (**c**) Integrated wound atlas UMAP. Dark color clusters are enriched in wound cells. We detect our populations α and β from the *egr* wound and two additional *ras^V12^scrib^-/-^* and *scrib^-/-^*

*Figure 4 continued on next page*

*Figure 4 continued*

wound populations. (**d**) Distribution of RNA measures in wound α and β gene signature; asterisks denote significant enrichment compared to wild-type. (**e**) Scatter plot of the mean expression fold-change of gene sets extracted from eGRN, GO, marker genes ($ras^{V12}scrib^{-/-}$ and $scrib^{-/-}$), and β-specific regions with CEBP or CEBP-TEAD motifs hits (target genes inferred from the CEBP eGRNS). Fold-changes are contrasting gene expression changes between (x) wound population α and β and (y) between wild-type pouch domain and wound α. All elements are significantly up- or downregulated in at least one axis (p-adj < 5*10e$^{-3}$, paired *t*-test, Bonferroni corrected).

The online version of this article includes the following figure supplement(s) for figure 4:

**Figure supplement 1.** ATAC, RNA, and TE signatures in wound-specific populations.

**Figure supplement 2.** Tumor sub-clustering analysis.

via a JNK/STAT regulatory feedback loop (*Jaiswal et al., 2022*). Our results suggest that the tumor and senescent populations de-differentiated from their original state (eye or wing discs) toward a similar senescent-like wound-response program, although we cannot clearly distinguish the specific activity of **α** and **β** signatures in distinct tumor clusters.

## Discussion

In this study, we used several wound paradigms from *Drosophila* imaginal discs, both transient (following normal wounding) and persistent (continuous wound induction during tumor formation), to describe the mechanisms coordinating stress response. We constructed eGRNs from single-cell ATAC and RNA data, assigning TFs to their inferred regulatory target genes and target enhancer regions. Using this new eGRN viewpoint, we found two distinct populations respectively displaying proliferative (**α**) and senescent (**β**) properties. Both populations share a common wound response program, comprising JNK and JAK/STAT signaling. Yet, the senescent cells display an additional activation of DNA damage, paracrine signaling, and stress mitigation pathways (glutathione metabolism, FC$_{α/β}$ > 1.8) that is distinct from the proliferative cells. In agreement with a possible de-differentiation process, TFs associated with pouch fate specification are also downregulated in senescent cells (*Distal-less, blistered, engrailed*). Concurrently, the senescence markers co-vary with C/EBP-like eGRN activity and the oncogenic signatures ($ras^{V12}scrib^{-/-}$ and $scrib^{-/-}$, top 100 target genes, *Figure 4e*). Taken together, these results highlight a common response program, shared by all wound populations and $ras^{V12}scrib^{-/-}$ cancer cells (JNK, JAK/STAT), and an additional senescent-like program driven by C/EBP-like TFs that is shared between the senescent population and the tumorigenesis paradigms. To our knowledge, this study presents the first depiction of the senescence phenotype at the single cell multiomic level (scRNA+scATAC). We believe our characterized gene and enhancer signatures will help pave the way toward a universal marker set of senescence, which is currently outstanding (*Cohn et al., 2023*; *Gorgoulis et al., 2019*). Using these single-cell senescent profiles along with their eGRN signatures, we can already show here a close proximity between the senescence phenotype and oncogenic programs, mostly due to the shared upregulation of C/EBP transcription factors activity.

The persistent wound paradigm (e.g. $ras^{V12}scrib^{-/-}$) causes uncontrolled neoplastic growth and has previously been referred to as a tumor model (*Dillard et al., 2021*; *Mirzoyan et al., 2019*). In this study, we found tumor programs to be highly similar to the senescent-like cell population emerging following a transient wound. These findings corroborate the growing set of evidence suggesting a large overlap between the hallmarks of tumor and wound response (*MacCarthy-Morrogh and Martin, 2020*). From this perspective, we can see a tumor paradigm as a wound that never heals, where the failure to arrest cell proliferation would prevent the damaged cells to fully resorb.

The senescent phenotype has been extensively studied, notably in the context of wound repair (*Demaria et al., 2014*; *Rhinn et al., 2019*; *Wilkinson and Hardman, 2020*). It is defined as a state of arrested cell proliferation, combined with a strong paracrine signaling activity (SASP) promoting stress-response signaling and partly mediated by CEBP/B activation. Recent studies further suggest a guiding role, where senescent cells lead the collective migration of proliferative cells toward wound closure, where they are then discarded through mechanical competition (*Álvarez-Fernández et al., 2015*; *Kozyrska et al., 2022*; *Tsai et al., 2018*).

The mechanism mediating the clearance of senescent cells is critical to prevent the persistence of stress signaling, as observed in tumor models (*Baumgartner et al., 2021*; *Kozyrska et al., 2022*; *Pinal et al., 2019*). In line with the hypothesis of competition-mediated clearance of senescent cells

(*Kozyrska et al., 2022*), recent studies developed the importance of C/EBP in stress response and cell competition, although no clear consensus currently exists regarding its type of action (promoting or inhibiting growth and survival) (*Baillon et al., 2018*; *Blanco et al., 2020*; *Boulan et al., 2019*; *Brown et al., 2021*; *Huggins et al., 2015*; *Lee et al., 2018*; *Logeay et al., 2022*). Given the ambivalent needs for senescent cells to firstly survive a strong apoptotic signal (ablation) and secondly get discarded through competition (closure), we believe that further investigations of the C/EBP-mediated regulation of apoptosis susceptibility will be of high interest. Following this hypothesis, failure to arrest SASP signaling through senescent cell clearance would promote overproliferation in persistent oncogenic induction. This idea of competition-mediated clearance is supported by several studies showing that polarity sensing mutants (e.g. *scrib*[-/-]) would overproliferate when present as a homogeneous population, but would be cleared within a crowd of wild-type cells (*Ballesteros-Arias et al., 2014*; *Brumby and Richardson, 2003*; *Pinal et al., 2019*).

The mechanism governing the sporadic emergence and persistence of senescent cells at the wound site is still unclear. Previously it was shown that the DNA damage-responsive TF p53 and cell cycle stalling cooperate to trigger cellular senescence (*Kozyrska et al., 2022*). Such stalling has also been reported in the *Drosophila* wound model, where it is mediated by AP-1 and involves *tribbles* (*Cosolo et al., 2019*), two markers of our senescent population. The authors further highlight that JNK activity is protecting the damaged, senescent cells from competition as long as they do not resume cycling. Two other studies have also demonstrated the role of a positive feedback loop via *moladietz* (marker of our proliferative population α) to sustain JNK activity in the senescent cells after wounding (*Khan et al., 2017*; *Pinal et al., 2018*). Additional studies suggest the action of microRNAs (*Bilak et al., 2014*) and transposable elements (TE) (*Azpiazu and Morata, 2022*; *Colombo et al., 2018*) to induce cell senescence via chromatin remodeling pathways. In line with this hypothesis, we find specific expression of the retrotransposons TAHRE and Invader4 in the wound populations (p-adj < $10e^{-3}$, log2FC > 1, *Figure 4—figure supplement 1c*), although further investigation would be necessary to fully characterize this TE-mediated response.

We do not detect a strong signal for the signature of several TFs that have been previously suggested to play a role in the wound response, such as p53. Indeed, we do not find evidence for the activity of p53 eGRN at the chromatin level (*Jacobs et al., 2018*), even in the senescent population where it is present as a C/EBP target gene (*Figure 3—figure supplement 3a*). This is also not the case for *Grainyhead*, which is involved in ERK signaling and epithelial barrier repair (*Mace et al., 2005*), but is not upregulated in the wound response state (*Figure 4—figure supplement 1d*). The Hippo pathway (pro-proliferative and pro-invasive in *ras*[V12]*scrib*[-/-]) is also not detected as a wound signature in our *rn-egr* model, although we find an enrichment of its effector's motif, Scalloped, in the senescent wound population. The presence of *sd* is consistent with the reported cooperation of its human ortholog, TEAD, with AP-1 in invasive mesenchymal-like cells (*Verfaillie et al., 2015*; *Zanconato et al., 2015*).

Based on the eGRN signatures, our results indicate that the wound population β, marked by high paracrine and DNA-damage signaling, corresponds to a senescent cell type. Our data supports a model in which these senescent cells are capable of surviving apoptotic signals by cell-cycle stalling, and then further guide the migration and growth of the pro-regenerative population (α) via its strong secretory phenotype, notably targets of the JNK pathway (i.e. presence of Wg and Upd family ligands; *La Marca and Richardson, 2020*). The significant similarity between senescent cells and *ras*[V12]*scrib*[-/-] cells suggests that they share the same regulatory wound programs, highlighting the fact that mechanisms present in oncogenic induction may already be active in a genetically wild-type context. Nevertheless, our current data do not capture clear cellular heterogeneity in *ras*[V12]*scrib*[-/-] model, as observed in the wound data, and are inconclusive of whether the α and β signatures are activated in distinct populations or in an aberrant combination in tumor cells. A recent study from *Jaiswal et al., 2022* observed different spatial patterns of markers at the wound site in the *ras*[V12]*scrib*[-/-] model. These populations arise after a persistent activity of a JNK/STAT regulatory network that stratifies signaling activity and cell behavior around the wound site, analogous to morphogen gradient fields in development. Their hypothesis of a wound organizer and supporting senescent marker stainings agrees with our observation of senescent cells being at the center of the wound, while representing a gradient of senescent marker activity in the *ras*[V12]*scrib*[-/-] tumor.

Given the preponderance of the C/EBP eGRNs we observe in tumor paradigms, we believe that further characterization of their signaling pathways and experimental validation of their predicted gene and enhancer targets will help to study the biology of cancer. Regarding TF cooperation events, multiple bZip TFs present in our study are conserved in the mammalian inflammatory pathways and are capable of forming heterodimers (*Blanco et al., 2020*; *Deppmann et al., 2006*; *Deppmann et al., 2006*; *Huggins et al., 2015*; *Newton and Dixit, 2012*; *Shokri et al., 2019*). Notably, a recent study in YAP/TAZ-bound regions upregulated in breast cancer has detected an increased binding co-occurrence for the orthologs of our key regulatory TFs: AP-1, STAT3, C/EBP, and TEAD (Jra, kay, Stat92E, Sd, Irbp18, slbo) (*He et al., 2021b*).

## Materials and methods

### Fly genotypes

The following fly stocks were used for ras$^{V12}$, scrib$^{-/-}$ experiments: y,w,eyFlp; act>y +> Gal4, UAS-GFP/ UAS-RasV12; FRT82 tub-Gal80/FRT82 scrib-,e. The fly stocks for *egr* experiments are w$^{1118}$; +; rn-Gal4, UAS-eiger, tub-Gal80$^{ts}$/TM6B. The fly stocks for Xrp1 experiments are w$^{1118}$; w; UAS-Xrp1-S/TM6B (*Boulan et al., 2019*); w; UAS-mCD8-ChRFP; rn-Gal4, tub-Gal80$^{ts}$/TM6B (*Smith-Bolton et al., 2009*, mCD8-ChRFP added in the Uhlirova Lab).

### Genetic cell ablation using Gal4/UAS/Gal80ts

To induce expression of *egr*, experiments were carried out as described in *Cosolo et al., 2019*; *La Fortezza et al., 2016*; *Smith-Bolton et al., 2009* with a few modifications. Briefly, larvae of genotype *rn-Gal4, tub-Gal80$^{ts}$* and carrying the desired *UAS*-transgenes were staged with a 6 hr egg collection and raised at 18°C at a density of 50 larvae/vial. Overexpression of the TNF ligand *egr* transgene was induced by shifting the temperature to 30°C for 24 hr at day 7 after egg deposition and larvae were dissected right after. Imaginal wing discs were collected from wandering third-instar larvae in PBS and flash-frozen in liquid nitrogen.

### Neoplastic growth induction using *ras$^{V12}$*, *scrib$^{-/-}$*

Flies were raised at 25°C on a yeast-based medium under a 12 hr–12 hr day–night light cycle. *ras$^{V12}$, scrib$^{-/-}$* early eye-antennal discs were dissected from wandering third-instar larvae (days 6–7) in PBS. *ras$^{V12}$, scrib$^{-/-}$* late were collected 4 d after larvae began wandering (days 10–11); this is possible because *ras$^{V12}$, scrib$^{-/-}$* do not pupate, but can persist more than 1 wk in a prolonged larval stage.

### Xrp1 overexpression

To induce expression of UAS-based transgenes, control (*rn$^{ts}$>; UAS-mCD8-ChRFP, rn-Gal4, tub-Gal80$^{ts}$*) and *rn$^{ts}$ >Xrp1S* (*UAS-mCD8-ChRFP, UAS-Xrp1-S, rn-Gal4, tub-Gal80$^{ts}$*) larvae were raised at 22°C and shifted to 29°C on day 7 AEL. Wing imaginal discs (120 WIDs for *rn$^{ts}$>*, 160 WIDs for *rn$^{ts}$ >Xrp1S*) were dissected 48 hr later in PBS and flash-frozen in liquid nitrogen. Total RNA was isolated according to standard TRI Reagent protocol (Sigma-Aldrich, #T9424), followed by DNase I treatment (Invitrogen, #AM2238) and repurification as described in *Mundorf and Uhlirova, 2016*. RNA-seq libraries were prepared according to TruSeq stranded mRNA sample preparation guide (Illumina).

### Immunohistochemistry

Wing discs from third-instar larvae were dissected and fixed for 15 min at room temperature in 4% paraformaldehyde in PBS. Washing steps were performed in PBS containing 0.1% TritonX-100 (PBT). Discs were then incubated with primary antibodies in PBT, gently mixing overnight at 4°C. Tissues were counterstained with DAPI (0.25 ng/μl, Sigma, D9542), Phalloidin-Alexa Fluor 488/647 (1:100, Life Technologies) or Phalloidin-conjugated TRITC (1:400, Sigma) during incubation with cross-absorbed secondary antibodies coupled to Alexa Fluorophores (Invitrogen or Abcam) at room temperature for 2 hr. The gstD-GFP is a GFP reporter under the control of 2.7 kB upstream regulatory region of GstD as published by Dirk Bohmann's lab (*Sykiotis and Bohmann, 2008*) and the Trbl GFP is a GFP-trap MIMIC line inserted in the intron of Trbl (Bloomington stock #61654). Tissues were mounted using SlowFade Gold Antifade (Invitrogen, S36936). Whenever possible, experimental and control discs were processed in the same vial and mounted on the same slides to ensure comparability in staining

between different genotypes. Images were acquired using the Leica TCS SP8 Microscope, using the same confocal settings and processed using tools in Fiji. Per-channel views are shown in *Figure 3—figure supplement 5*.

## Sample and library preparation for single-cell gene expression

### Sample preparation

Eye-antennal discs or wing discs were dissected and transferred to a tube containing 100 μl ice-cold PBS. After centrifugation at $800 \times g$ for 5 min, the supernatant was replaced by 50 μl of dispase (3 mg/ml, Sigma-Aldrich_D4818-2mg) and 75 μl collagenase I (100 mg/ml, Invitrogen_17100-017). Discs were dissociated at 25°C in a Thermoshaker (Grant Bio PCMT) for 45 min at 25°C, 500 rpm. The enzymatic reaction was reinforced by pipette mixing every 15 min. Cells were washed with 1 ml ice-cold PBS and resuspended in 400 μl PBS supplemented with 0.04% BSA. Cell suspensions were passed through a 10 μM pluriStrainer (ImTec Diagnostics-435001050). Cell viability and concentration were assessed by the LUNA-FL Dual Fluorescence Cell Counter.

### Library preparation

Single-cell libraries were generated using the 10X Chromium Single-Cell Instrument and Single Cell 3' Gene Expression (GEX) kit according to the manufacturer's protocol. Briefly, single cells from eye-antennal discs or wing discs were suspended in 0.04% BSA-PBS. After generation of nanoliter-scale Gel bead-in-emulsions (GEMs), GEMs were reverse transcribed in a C1000 Touch Thermal Cycler (Bio-Rad) programmed at 53°C for 45 min, 85°C for 5 min, and hold at 4°C. After reverse transcription, single-cell droplets were broken and the single-strand cDNA was isolated and cleaned with Cleanup Mix containing DynaBeads (Thermo Fisher Scientific). cDNA was then amplified by PCR: 98°C for 3 min; 12 cycles of 98°C for 15 s, 67°C for 20 s, 72°C for 1 min; 72°C for 1 min; and hold at 4°C. Subsequently, the amplified cDNA was fragmented, end-repaired, A-tailed and index adaptor ligated, with SPRIselect cleanup in between steps. The final gene expression library was amplified by PCR: 98°C for 45 s; 14 cycles of 98°C for 20 s, 54°C for 30 s, 72°C for 20 s; 72°C for 1 min; and hold at 4°C. The sequencing-ready library was cleaned up with SPRIselect beads.

### Sequencing

Before sequencing, the fragment size of every library was analyzed using the Bioanalyzer high-sensitivity chip. All 10× GEX libraries were sequenced HiSeq4000 or NovaSeq6000 instruments (Illumina) with the following sequencing parameters: 26 bp read 1–8 bp index 1 (i7) – 98 or 75 bp read 2.

## Sample and library preparation for 10× single-nuclei multiome ATAC and gene expression

### Sample preparation

Control and wounded wing discs were dissected and transferred to a tube containing ice-cold PBS. PBS was removed by centrifugation, tissues were flash frozen in liquid nitrogen and stored at –80°C. The following procedure was followed to extract the nuclei from the wing discs: resuspension in 500 μl nuclei lysis buffer comprising 10 mM Tris-HCl pH 7.4, 10 mM NaCl, 3 mM MgCl$_2$, 0.1% Tween-20, 0.1% Nonidet P40, 0.01% Digitonin, 1% BSA, 1 mM dithiothreitol, and 1 U/μl RNasin ribonuclease inhibitor (Promega) in nuclease-free water, incubation on ice for 5 min, transfer to a dounce tissue grinder tube (Merck), 25 strokes with pestle A, incubation on ice for 10 min, 25 strokes with pestle B. The lysis was stopped by added 1 ml of wash buffer composed of 10 mM Tris-HCl pH 7.4, 10 mM NaCl, 3 mM MgCl$_2$, 0.1% Tween 20, 1% BSA, 1 mM dithiothreitol, and 1 U/μl RNasin ribonuclease inhibitor in nuclease-free water. Nuclei were pelleted by centrifugation at $800 \times g$ for 5 min at 4°C and resuspended in a 1× nuclei buffer (10X Genomics) supplemented with 1 mM dithiothreitol and 1 U/μl RNasin ribonuclease inhibitor. Nuclei suspensions were passed through a 40 μm Flowmi filter (VWR Bel-Art SP Scienceware). Nuclei concentration was assessed using the LUNA-FL Dual Fluorescence Cell Counter.

## Library preparation

Single-cell libraries were generated using the 10X Chromium Single-Cell Instrument and NextGEM Single Cell Multiome ATAC+Gene Expression kit (10X Genomics) according to the manufacturer's protocol. In brief, the single nuclei of wing discs were incubated for 60 min at 37°C with a transposase that fragments the DNA in open regions of the chromatin and adds adapter sequences to the ends of the DNA fragments. After generation of nanoliter-scale gel bead-in-emulsions (GEMs), GEMs were incubated in a C1000 Touch Thermal Cycler (Bio-Rad) under the following program: 37°C for 45 min, 25°C for 30 min, and hold at 4°C. Incubation of the GEMs produced 10× barcoded DNA from the transposed DNA (for ATAC) and 10× barcoded, full-length cDNA from poly-adenylated mRNA (for GEX). This was followed by a quenching step that stopped the reaction. After quenching, single-cell droplets were broken and the transposed DNA and full-length cDNA were isolated using Cleanup Mix containing Silane Dynabeads. To fill gaps and generate sufficient mass for library construction, the transposed DNA and cDNA were amplified via PCR: 72°C for 5 min; 98°C for 3 min; seven cycles of 98°C for 20 s, 63°C for 30 s, 72°C for 1 min; 72°C for 1 min; and hold at 4°C. The pre-amplified product was used as input for both ATAC library construction and cDNA amplification for gene expression library construction. Illumina P7 sequence and a sample index were added to the single-strand DNA during ATAC library construction via PCR: 98°C for 45 s; 7–9 cycles of 98°C for 20 s, 67°C for 30 s, 72°C for 20 s; 72°C for 1 min; and hold at 4°C. The sequencing-ready ATAC library was cleaned up with SPRIselect beads (Beckman Coulter). Barcoded, full-length pre-amplified cDNA was further amplified via PCR: 98°C for 3 min; 6–9 cycles of 98°C for 15 s, 63°C for 20 s, 72°C for 1 min; 72°C for 1 min; and hold at 4°C. Subsequently, the amplified cDNA was fragmented, end-repaired, A-tailed, and index adaptor ligated, with SPRIselect cleanup in between steps. The final gene expression library was amplified by PCR: 98°C for 45 s; 5–16 cycles of 98°C for 20 s, 54°C for 30 s, 72°C for 20 s, 72°C for 1 min; and hold at 4°C. The sequencing-ready GEX library was cleaned up with SPRIselect beads.

## Sequencing

Before sequencing, the fragment size of every library was analyzed using the Bioanalyzer high-sensitivity chip. All 10× Multiome ATAC libraries were sequenced on NovaSeq6000 instruments (Illumina) with the following sequencing parameters: 50 bp read 1–8 bp index 1 (i7) – 16 bp index 2 (i5) – 49 bp read 2. All 10× Multiome GEX libraries were sequenced on NovaSeq6000 instruments with the following sequencing parameters: 28 bp read 1–10 bp index 1 (i7) – 10 bp index 2 (i5) – 75 bp read 2.

## scATAC and scRNA read mapping

All the analyses performed in this study used the *Drosophila melanogaster* r6.35 (dm6) reference sequence and annotations. All single-cell datasets were (re)processed using CellRangerARC/1.0.1 and CellRanger/5.0.1 with default parameters for multiome and scRNA, respectively.

## scRNA analysis

We analyzed the entire dataset with an in-house-developed NextFlow pipeline, named VSN (*Flerin et al., 2021*). VSN performs a standard scRNA analysis with *best-practices* workflow (*Luecken and Theis, 2019*), including the doublet filtering using *scrublet* (*Wolock et al., 2019*) and the correction for batch effects between the runs using *Harmony* (*Korsunsky et al., 2019*; *Figure 1—figure supplement 2a*). scRNA data were processed with default parameters: minimum of 200 detected genes per cell, maximum of 15% signal of mitochondrial origin, minimum of 3 cells expressing any targeted genes. We additionally introduced the quality-check step from *Everetts et al., 2021* to filter for low-quality cell abundance in the integrative analysis combining our runs together with public datasets. This filtering step consists in removing cell clusters with a mean number of detected genes lower than 1 SD below the global average. The clustering, UMAP, and tSNE resulting from the integrative analysis were computed on 50 principal components (PCs, selected with the *pcacv* module; *Varmuza and Filzmoser, 2009*) and a Leiden resolutions of 0.5 and 1.4 for the wild-type and wound (*egr, ras^{V12}, scrib^{-/-}*) atlas integrations, respectively. For the multiome-only analysis, clustering and wild-type annotations were derived from the wild-type integrated analysis. UMAP and tSNE were computed on 50 PCs. Detection of the wound subpopulations α and β was done on the same PC space, at a Leiden resolution of 1.3. Marker genes were detected on log-normalized counts using the standard *scanpy*

*rank_genes_groups* module with default or more stringent parameters (**Wolf et al., 2018**; method = *t-test_overestim_var*, detailed values in the 'Results' section). Analysis of Transposable Element signal was done on the multiome data using scTE (**He et al., 2021a**) with identical preprocessing steps.

## scATAC analysis

The analysis of chromatin profiles obtained from our three multiome runs was done using cisTopic (**Bravo González-Blas et al., 2019**). One critical step of this analysis is to confidently identify the *cis*-regulatory regions active in our system. To do so, we generated pseudo-bulk ATAC tracks by aggregating the sequencing outputs from cells with identical scRNA annotation together. For each track, we then define peaks as 500 bp regions centered around MACS2-called summits (**Zhang et al., 2008a**). The consensus peak set (CPS) of 41,387 regions, additionally filtered for repeat loci, is constructed via the Iterative Overlap Peak Merging Procedure (**Corces et al., 2018**). In addition to the scRNA-based cell filtering (see 'scRNA analysis' section), we filter the cells for three scATAC-based metrics: minimum number of fragment per cell of $\log_{10}(3.5)$, minimum fraction of reads in CPS of 60%, minimum TSS enrichment of 2. Using the filter cell and the CPS to generate an input count matrix, we run cisTopic LDA and select the model with best topic coherence (n_topic = 76, **Figure 1—figure supplement 2e**). We use the Harmony-corrected topic dimensions to compute tSNE and UMAP embeddings (**Figure 1—figure supplement 2f**), which recapitulate the main scRNA cell types, despite a noticeable mixing of some clusters (e.g. direct and indirect AMPs are mixed and will be merged as a single annotation in the following analyses). The imputed accessibility scores are computed from the 76-topic model per cell for each detected enhancer region. This probability matrix serves as a proxy for chromatin accessibility in the GRN inference analysis, while limiting the impact of dropout values (see 'Materials and methods'). The DARs are computed for each cell type and conditions using a Wilcoxon rank-sum test (p-adj<5 * 10e$^{-3}$, log2FC > 1.5, mean 1770 DARs per cell type).

## (pseudo)bulk ATAC analysis

The wound population β does not form a clear cluster in the scATAC embeddings (**Figure 1—figure supplement 2d**), which might indicate a non-optimal detection of their accessibility profile in the cistopic model (potentially due to low cell number). In order to maximize the detection of the accessibility profiles of wound population α and β, we took advantage of the ATAC pseudo-bulk tracks generated previously and directly computed the DARs between **α** and **β** using the bdgdiff module from MACS2. We find 3980 regions accessible in both populations, 338 **α**-specific and 173 **β**-specific DARs (log-likelihood ratio > 1.5, maximum gap = 150, minimum length = 300). The feature maps presented in **Figure 3b** and **Figure 3—figure supplement 3a** were obtained by scoring a group of six TF motifs on each of these regions using cluster-buster (**Frith et al., 2003**) with a minimum motif score of 6. Detailed region coordinates along with motif hit information can be found in the **Supplementary file 6**.

The bulk ATAC data from wild-type and *ras^V12^scrib^-/-^* eye-antennal discs was reprocessed similarly to the original study (**Davie et al., 2015**) with updated reference genome (dm6). Reads were mapped with Bowtie2, regions were called using MACS2 and DARs were detected via DESeq2. We obtain a signature of 4547 regions significantly more accessible in *ras^V12^,scrib^-/-^* conditions compared to wild-type (p-adj<5*10e$^{-3}$, log2FC > 1.5). This signature overlaps 3236 regions from the CPS.

## Bulk RNA analysis

Xrp1 overexpression data was mapped using STAR with default parameters and no multimapping reads (outFilterMultimapNmax = 1) on dm6 r6.35 reference genome (sjdbOverhang = 99, genomeSAindexNbases = 12). Gene counts were obtained using htseq-count with default parameters, and differential analysis was performed using DESeq2 (logFC > 1.5, p-adj < 5.10e$^{-3}$). GSEA was performed with the R package {fgsea} on log2 fold-change ranking from all detected genes (4991 genes, default filtering on mean normalized count). Cellular senescence gene sets were extracted from ReactomeDB (R-DME-2559583), and full set and individual subsets were tested for enrichment.

## eGRN inference

The computational pipeline used to infer eGRN is principally based on previous works from **Bravo González-Blas et al., 2020** and **Janssens et al., 2022** and can be summarized in three steps

schematized in *Figure 2a*. In the first step, we apply *cistarget* to each of the DARs computed in the scATAC analysis (wild-type and wound, see 'Materials and methods') as well as the pseudo-bulk contrast between wound populations **α** and **β**. We retain TF hits in each region of a set of DARs if the TF motif is overall significantly enriched in the set (NES > 3, rank threshold = 0.05, AUC threshold = 0.005, motif similarity FDR = 0.001, orthologous identity threshold = 0.6) and the motif is actually detected as a hit in the region of interest. The scoring database used for this step corresponds to the cisTarget motifs collection v9 (28,799 motifs), specifically re-scored on the CPS from this study (instead of the standard 131,324 candidate regulatory regions for fly i-cisTarget). To avoid issues with redundant motifs within the database (e.g. 14 motifs are associated with human JUND), we grouped similar motifs into a merged consensus prior to scoring. The final set of regions targeted by a specific TF is called a TF's cistrome. We obtained two categories of cistromes for each TF: a direct cistrome, solely retaining results from motifs with direct TF annotation, and an extended cistrome, including enrichment results from motifs passing the similarity or orthology threshold.

In the second step, we use the log-normalized gene expression and imputed accessibility signal from our multiome runs as input (for preprocessing, see 'Materials and methods'). Using both views, we can calculate region-to-gene relationship using a nonlinear regression method. We compute both importance (Gradient Boosting Regression with arboreto) and correlation (Spearman) scores for all region-gene pairs having a distance between the region and the TSS within the range of 100–50k base pairs. In the third step, we extract similar importance and correlation scores as in step 2, this time contrasting the gene expression values between the TFs and target genes. For this step, we directly use the output derived from pyscenic analysis with default parameters included in the VSN pipeline (*Van de Sande et al., 2020*).

Last but not least, the critical step required to finalize the eGRN inference is to prioritize the edges characterized in previous steps that have the highest probability to be functional. First, we dichotomize all edges into activating or repressing functional categories based on a Spearman's $\rho$ threshold of 0.03 (activating if $\rho$ > 0.03, repressing if $\rho$ < 0.03, other edges are discarded). Similar to the method from *Aibar et al., 2017*, we then generate sets of refined region-to-gene edges via multiple pruning methods based on importance scores. The methods used in this analysis are BASC binarization (*Hopfensitz et al., 2012*), quantiles (top 0.75, 0.80, 0.85, 0.90, and 0.95 quantiles for all regions associated with a given gene) and top highest importance (top 5, 10, and 15 regions associated with a given gene). From these sets of regions, we include specific TF information by associating a TF-to-region edge to each region present in one or multiple TF's cistrome, from both direct and extended categories (see *cistaret* analysis in step 1).

The fourth and last step of the GRN inference consists of pruning the resulting list of TF-to-region-to-gene edges based on the TF-to-gene importance ranking obtained from step 3. To do so, we retain the leading edges from a GSEA (*Subramanian et al., 2005*), taking the TF-to-gene importance from step 3 as ranking values and the target gene from step 1–2 as the tested gene set. Lastly, we unified all the leading region-to-gene edges from the same cistromes and functional category into a common eGRN. This way, for each TF, we preserve the distinction between activating, repressing, direct and extended eGRNs.

## eGRN analysis

Having access to eGRN information allows us to further explore our multiome dataset. We can score an eGRN activity at the single-cell level for both gene expression and chromatin accessibility using the *AUCell* module from the SCENIC pipeline. An eGRN will have a high AUC score in a cell if its set of target genes and regions are overall ranking high in terms of gene expression and chromatin accessibility, respectively. To limit the impact of population size, we compute AUC scores on 150 pseudo-bulk cells from each cell type, each generated from a random selection of 15 cells for the same cell type. We define a set of high-quality eGRN by computing Pearson's correlation between TF expression and eGRN activity across the pseudo-bulk cells and retain the regulons with an absolute $\rho$ higher than 0.2. We additionally remove extended eGRN from the selection if the direct eGRN from the same TF is already present. The final list of eGRN comprises 98 TFs and 147 eGRNs, and is available in *Supplementary file 2*, along with correlation results.

In addition to pseudo-bulk, we also derived AUC scores from the full multiome dataset at single-cell level. These scores can be queried via the SCope platform (https://scope.aertslab.org/#/WingAtlas/*/

welcome) and are shown as an example in *Figure 2b*. Using these AUC scores, we used the method presented in *Suo et al., 2018* to compute an RSS for each eGRN. This RSS indicates whether the eGRN target gene and region sets are specifically active in one cell type (*Figure 2—figure supplement 1a*). We retrieve the top-scoring eGRN for each cell type to build the *Figure 2c* (gene-based RSS) and *Figure 3a* (region-based RSS, NES scores retrieved from the cistarget analysis).

## Acknowledgements

This work was funded by the following grants to SA: ERC Consolidator Grant (724226_cisCONTROL), Special Research Fund (BOF) KU Leuven (grant C14/18/092), and FWO-Vlaanderen (G0C0417N, G094121N); the grants to AC from the Deutsche Forschungsgemeinschaft (DFG, German Research Foundation) under Germany's Excellence Strategy (CIBSS – EXC-2189 – Project ID 390939984), the Heisenberg Program (CL490/3-1) and by the Boehringer Ingelheim Foundation (Plus3 Programme); the grant to MU under Germany's Excellence Strategy (CECAD, EXC 2030 – 390661388) from the Deutsche Forschungsgemeinschaft (DFG, German Research Foundation); MA is supported by the Roland Black Endowed Assistant Professorship from Sam Houston State University.

## Additional information

### Funding

| Funder | Grant reference number | Author |
|---|---|---|
| European Research Council | 724226_cisCONTROL | Valerie M Christiaens<br>Gert J Hulselmans |
| Fonds Wetenschappelijk Onderzoek | G0C0417N | Xiaojiang Quan<br>Duygu Koldere |
| Fonds Wetenschappelijk Onderzoek | G094121N | Swann Floc'hlay |
| Deutsche Forschungsgemeinschaft | EXC 2189 CIBSS | Anne-Kathrin Classen |
| Deutsche Forschungsgemeinschaft | CL490/3-1 | Anne-Kathrin Classen |
| Deutsche Forschungsgemeinschaft | EXC 2030 | Mirka Uhlirova |
| KU Leuven | C14/18/092 | Maxime De Waegeneer |
| Boehringer Ingelheim Foundation | Plus3 Programme | Anne-Kathrin Classen |

The funders had no role in study design, data collection and interpretation, or the decision to submit the work for publication.

### Author contributions

Swann Floc'hlay, Data curation, Formal analysis, Validation, Investigation, Visualization, Methodology, Writing – original draft; Ramya Balaji, Resources, Methodology; Dimitrije Stanković, Validation; Valerie M Christiaens, Xiaojiang Quan, Methodology; Carmen Bravo González-Blas, Data curation, Software, Formal analysis, Methodology; Seppe De Winter, Formal analysis, Methodology; Gert J Hulselmans, Maxime De Waegeneer, Software; Duygu Koldere, Conceptualization, Formal analysis, Methodology; Mardelle Atkins, Resources, Supervision, Writing – review and editing; Georg Halder, Conceptualization, Resources, Supervision, Writing – review and editing; Mirka Uhlirova, Validation, Writing – review and editing; Anne-Kathrin Classen, Conceptualization, Resources, Supervision, Funding acquisition, Methodology, Writing – review and editing; Stein Aerts, Conceptualization, Data curation, Supervision, Funding acquisition, Investigation, Methodology, Writing – original draft, Project administration, Writing – review and editing

### Author ORCIDs

Swann Floc'hlay  http://orcid.org/0000-0003-1477-830X
Seppe De Winter  http://orcid.org/0000-0001-7907-1247
Mardelle Atkins  http://orcid.org/0000-0002-0245-2452
Mirka Uhlirova  http://orcid.org/0000-0002-5735-8287
Anne-Kathrin Classen  http://orcid.org/0000-0001-5157-0749
Stein Aerts  http://orcid.org/0000-0002-8006-0315

### Decision letter and Author response

Decision letter https://doi.org/10.7554/eLife.81173.sa1
Author response https://doi.org/10.7554/eLife.81173.sa2

## Additional files

### Supplementary files

• Supplementary file 1. Markers of the atlas cluster (leiden_resolution = 0.5), using scanpy rank_genes_groups, p_adj cutoff 0.01, log2FC cutoff of log2(3).

• Supplementary file 2. eRegulon metadata, orthology > 0.6, no min target set size, keep ext. if no direct, 0.2 minimum rho.

• Supplementary file 3. Enhancer DARs accessible in wound population with overlap information from previous work from Harris et al (EL3DR) and Vizcaya-Molina et al (iDRRE).

• Supplementary file 4. Markers of the wound populations (alpha vs. beta and both versus pouch) using scanpy rank_genes_groups, p_adj cutoff 0.01, log2FC cutoff of log2(2).

• Supplementary file 5. Markers of the tumors subclusters (each vs. all tumor), using scanpy rank_genes_groups, p_adj cutoff 0.01, log2FC cutoff of log2(2).

• Supplementary file 6. Regions and TF motif hit information used to construct featuremaps include the CEBP-specific target genes.

• MDAR checklist

### Data availability

Single-cell sequencing data and aligned matrices have been deposited in GEO (accession code GSE205401). Single-cell gene expression and chromatin accessibility patterns for all analyses are accessible through the UCSC Genome Browser and through the SCope platform, along with eGRNs and their associated activity scores and target features.

The following dataset was generated:

| Author(s) | Year | Dataset title | Dataset URL | Database and Identifier |
|---|---|---|---|---|
| Floc'hlay S, Aerts S | 2022 | Shared enhancer gene regulatory networks between wound and oncogenic programs | https://www.ncbi.nlm.nih.gov/geo/query/acc.cgi?acc=GSE205401 | NCBI Gene Expression Omnibus, GSE205401 |

The following previously published datasets were used:

| Author(s) | Year | Dataset title | Dataset URL | Database and Identifier |
|---|---|---|---|---|
| Hariharan IK | 2021 | Single-cell transcriptomics of the *Drosophila* wing disc reveals instructive epithelium-to-myoblast interactions | https://www.ncbi.nlm.nih.gov/geo/query/acc.cgi?acc=GSE155543 | NCBI Gene Expression Omnibus, GSE155543 |
| Yan Y | 2019 | Single-cell transcriptomic analysis of 96hour AEL wild type wing imaginal discs | https://www.ncbi.nlm.nih.gov/geo/query/acc.cgi?acc=GSE133204 | NCBI Gene Expression Omnibus, GSE133204 |

*Continued on next page*

*Continued*

| Author(s) | Year | Dataset title | Dataset URL | Database and Identifier |
|-----------|------|---------------|-------------|-------------------------|
| Valentini E | 2019 | Gene expression atlas of a developing tissue by single cell expression correlation analysis | https://www.ncbi.nlm.nih.gov/geo/query/acc.cgi?acc=GSE127832 | NCBI Gene Expression Omnibus, GSE127832 |
| González-Blas CB, Aerts S | 2020 | Identification of genomic enhancers through spatial integration of single-cell transcriptomics and epigenomics | https://www.ncbi.nlm.nih.gov/geo/query/acc.cgi?acc=GSE141590 | NCBI Gene Expression Omnibus, GSE141590 |
| Yan Y | 2019 | Single-cell transcriptomic analysis of the scrib mutant wing imaginal discs | https://www.ncbi.nlm.nih.gov/geo/query/acc.cgi?acc=GSE130566 | NCBI Gene Expression Omnibus, GSE130566 |
| Aerts S | 2015 | Discovery of Transcription Factors and Regulatory Regions Driving In Vivo Tumor Development by ATAC-seq and FAIRE-seq Open Chromatin Profiling | https://www.ncbi.nlm.nih.gov/geo/query/acc.cgi?acc=GSE59078 | NCBI Gene Expression Omnibus, GSE59078 |
| Worley et al | 2022 | Ets21C sustains a pro-regenerative transcriptional program in blastema cells of *Drosophila* imaginal discs | https://www.ncbi.nlm.nih.gov/geo/query/acc.cgi?acc=GSE174326 | NCBI Gene Expression Omnibus, GSE174326 |

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
