## [Editor Report]

This study is an important progression in our understanding of wounding response and its relationship to malignancy. Although this topic has been previously addressed in genetic studies, the use of a systems biology approach here provides compelling support for the dual use of regulatory sequences to achieve context dependence for two linked but non-redundant tasks. Investigators in the fields of gene regulation, developmental biology as well as basic cancer research will find this manuscript to be both important and useful.

---

## [Decision Letter]

**Decision letter after peer review:**

Thank you for sending your article entitled "Shared enhancer gene regulatory networks between wound and oncogenic programs" for peer review at *eLife*. Your article is being evaluated by 2 peer reviewers, and the evaluation is being overseen by Utpal Banerjee as the Senior Editor. The reviewers have opted to remain anonymous

Please take a careful look at the comments from the reviewers. As you will see, Reviewer 2's comments are probably easier to respond to (except for the wing vs eye disc origin question), but reviewer 3 considers the work preliminary and more importantly, points to a "complete lack of validation". This last issue could take months to resolve if you were to consider new experiments using mutants or markers.

Finally, I completely agree with the reviewers that there are serious omissions in the citation of prior work. The Wnt6/Wnt1 shared enhancers, and particularly, a mention of and comparison with Hariharan's work that is pointed out by both reviewers is a major concern. Please check the paper for any others that may or may not have been pointed out.

*Reviewer #1:*

Understanding the commonalities between gene regulatory signatures during the wound response and cancer is highly relevant for the study of cancer and regenerative biology. As such, this work will be of interest to a broad audience. Although the authors do not delve into functional validation of the GRNs they identify, providing a comprehensive integrated dataset encompassing data from several labs will be valuable for the field. Furthermore, this is the first report of single nucleus multiomic analysis of the wound response in the fly system, which allows the authors to devise a very useful workflow for mapping GRNs. We would therefore support publication in *eLife*, provided a few technical points are addressed and some corrections are made to the manuscript.

1. Lines 395-4, the authors state: "Yet, unlike the wound response, we do not observe two distinct proliferative-like and senescent-like cell populations in tumors."

However, the cells in the tumor model (eye) are of a different origin than the regenerative dataset (wing). Can they exclude that the difference is due to the different origin of the cells?

2. The Hariharan lab recently published a dataset using the same regeneration model, at a similar time point (40h versus 44h after eiger induction as far as we can see). In this manuscript, Worley et al. identify two populations of wound-specific cells, termed Blastema 1/2. To add value to their own work, the authors should integrate this dataset with their own and directly comment whether their clusters ⍺ and β correspond to Blastema 2 and 1, respectively, as seems likely to be the case. As the current manuscript's β cluster is only composed of 94 cells, merging the datasets will also strengthen the analysis of this cluster. We don't think the authors need to be concerned at the overlap between the two datasets, as the focus of the manuscripts is different. Discussing the other manuscript and merging the datasets will be useful to the community.

3. In Figure 4d, both the wound ⍺ and β populations appear to have an elevated proliferative signature compared to control pouch cells, but the authors state on line 289: "In contrast, cells from cluster β do not show clear proliferative nor regeneration markers."

Can they resolve this apparent discrepancy?

4. The authors should explain in the results what the left of Figure 4b is meant to illustrate. In general, this section is quite dense and difficult to follow, and would benefit from some editing.

5. Lines 303-314: the authors list several SASP markers. As this is an important part of their work, could they produce a plot showing the increase of these markers in cluster β compared with the other populations? Also, it would be good to validate at least one of these markers by staining or HCR on injured versus control wing discs.

*Reviewer #2:*

The strengths of the manuscript include the generation of large-scale single-cell data sets (including single-cell transcriptomics and scRNAseq+sc-ATAC-seq), and the comparative analysis of the genes and eGRNs utilized during wound healing and tumorigenesis. The authors identify enhancers that are cell type specific through sc-ATAC-seq. Based on known transcription factor binding motifs, the authors find an enrichment for binding sites from AP-1 and STAT in enhancers utilized during both wound healing and tumorigenesis.

However, there are major weaknesses, including that the manuscript covers a large number of bioinformatic comparisons without almost any experimental validation within the wound or tumor models. In the few incidences where the authors show gene expression patterns within the wounded tissues, the images are not very convincing to identify the unique cell populations, and experimental validation is completely lacking from the tumor section. At multiple points in the paper, showing experimental evidence of the unique cell populations or enhancers within the wounded and/or tumor models would significantly advance the manuscript. In addition, the manuscript needs to improve placing the findings in a larger context within the field.

The comparisons between wounds and tumors that are made in this manuscript are of broad interest but many of the findings are still preliminary.

1. The analysis of the wounded imaginal wing discs does show a consistent location of wound α and wound β cells. Where are these cells located? Trbl-GFP (Figure 3d) expression appears to be elevated in a ring around the wound, while upd3 (Figure 1d) is in the center of the wound. In addition, GstD9-GFP appears to be expressed within a similar number of cells as MMP1, which is not consistent with the dot plot on the right side of these images. One of the claims of the paper is that they have characterized senescent markers – however, it is not clear where these cells are located within the wounded model and no localization within the tumor models is shown.

2. The word "senescent", which is used throughout the manuscript to describe the wound-β cells, is intriguing but ultimately potentially misleading and preliminary. Senescent implies that a cell will never enter the cell cycle again. The EDU experiment shows a reduction of cells in S-phase within the center of the wound. However, this is not quantified, and the positions of the wound-α and wound-β cells are not shown. In addition, the use of the word senescent cellular state to describe a growing tumor model is also confusing, especially when the authors conclude that the cells are homogenous. If the cells of the tumor are homogenous and the tumor is growing through cell proliferation, then cells with an activated senescent-cellular program must also be actively dividing.

3. Along similar lines, the paper strongly emphasizes the relationship between the cellular states activated within the wound and the cellular states within a tumor model. The major hypothesis from the bioinformatics should be investigated back in the tumor model. Are the two identified pathways (*JNK*/AP-1 and STAT) activated uniformly within the tumor model? If this GRN activates a cellular senescent-like program, and the cells within the tumors are in fact homogenous, then what mechanisms are driving the growth of the tumor? Recent work using single-cell data has highlighted the heterogeneity within scrib imaginal-disc tumors (eg. Deng et al., 2019; Ji et al., 2019; Worley et al., 2022). What do you think explains this difference between "scrib" vs "rasV12+scrib" cells? Are "rasV12+scrib" more homogenous than scrib alone? This statement would be greatly strengthened by visualizing co-expression of the "proliferative" or "senescent-like" pathways within these tumors to provide evidence that the rasV12 scrib-/- tumors are in fact co-activating and that the cells are homogenous.

4. The manuscript does not properly acknowledge previous work on damage-responsive enhancers (e.g. the wg/Wnt6 enhancer described in Harris et al. 2016; and the enhancers identified using Bulk-ATAC-seq by Vizcaya-Molina et al. 2018 and Harris et al. 2020). How do the enhancers uncovered in this paper relate to the enhancers uncovered in this past work? For a resource paper, it would be valuable to make stronger connections to how this work connects to prior knowledge.

5. Several points of this paper are similar to a single-cell study of regenerating imaginal discs that was recently published (Worley et al., 2022), including the comparison between wounded and developing tissues, as well as the comparison of damaged and tumorous tissues. How do the wound-α and wound-β relate to the identified blastema-1 and blastema-2 cell populations?

6. The statement that the senescent eGRN is activated and driven by a group of transcription factors needs additional experimental evidence. Driven applies that one or more of these transcription factors (Irbp18, Xrp1, etc.) would be required for the activation of these enhancers, and/or that the overexpression of one of these transcription factors would be sufficient. Experimental evidence is currently absent.

---

## [Author Response]

Reviewer #1:Understanding the commonalities between gene regulatory signatures during the wound response and cancer is highly relevant for the study of cancer and regenerative biology. As such, this work will be of interest to a broad audience. Although the authors do not delve into functional validation of the GRNs they identify, providing a comprehensive integrated dataset encompassing data from several labs will be valuable for the field. Furthermore, this is the first report of single nucleus multiomic analysis of the wound response in the fly system, which allows the authors to devise a very useful workflow for mapping GRNs. We would therefore support publication in eLife, provided a few technical points are addressed and some corrections are made to the manuscript.1. Lines 395-4, the authors state: "Yet, unlike the wound response, we do not observe two distinct proliferative-like and senescent-like cell populations in tumors."However, the cells in the tumor model (eye) are of a different origin than the regenerative dataset (wing). Can they exclude that the difference is due to the different origin of the cells?

The reviewer raises two points, firstly on the potential effect of cell origin (eye-antennal vs wing disc) and secondly on the lack of α/β separation (i.e. heterogeneity) in tumours. We will discuss them below separately.

Concerning the cell-of-origin effect, we added the following comment in the manuscript to provide clarity to the reader (line 391-395) : In this context, we believe the cell-oforigin has minor to no effect, given previous RNA-seq experiments performed in ras^V12^scrib^-/-^ eye, antennal, wing, and leg discs, that showed strong overlap of ectopically expressed genes (Atkins et al., 2016). This was also confirmed by recent in vivo experiments showing similar marker gene expression in the ras^V12^scrib^-/-^ models of both eye-antennal and wing imaginal discs (Jaiswal et al., 2022).Concerning the cellular heterogeneity, we have further explored our data within a new sub-clustering analysis. In particular, we performed a new integration of the *ras^V12^scrib**^-/-^*** scRNA-seq data datasets, specifically focusing on the tumor cells (i.e. excluding wing cells outside of the womb) to detect any potential sub-clustering patterns, and compare the candidate tumor sub-clusters to our α/β wound populations.(line 420-434, figure 4 – —figure supplement 2 and supplementary table 5). This sub-clustering analysis highlighted the presence of an expression gradient for several markers genes of senescence and stress response in tumor samples. In light of these results, we have nuanced our statement on the lack of cellular heterogeneity in the tumor (line 420-434 in results, line 545-555 in discussion). Lastly, we have also completed our conclusion with the perspective of the findings from Jaiswal et al. 2022, who demonstrate that cellular heterogeneity can be driven by a JAK/STAT versus *JNK* negative feedback (same paragraph reference).

2. The Hariharan lab recently published a dataset using the same regeneration model, at a similar time point (40h versus 44h after eiger induction as far as we can see). In this manuscript, Worley et al. identify two populations of wound-specific cells, termed Blastema 1/2. To add value to their own work, the authors should integrate this dataset with their own and directly comment whether their clusters ⍺ and β correspond to Blastema 2 and 1, respectively, as seems likely to be the case. As the current manuscript's β cluster is only composed of 94 cells, merging the datasets will also strengthen the analysis of this cluster. We don't think the authors need to be concerned at the overlap between the two datasets, as the focus of the manuscripts is different. Discussing the other manuscript and merging the datasets will be useful to the community.

We thank the reviewer for this suggestion, the dataset from Worley et al. is indeed highly valuable to compare with our data. We have performed an integrative ana;ysis of the scRNA data from our wound model and the two replicates from Worley et al. (line 291-294), providing a summary figure of the overlapping gene expression pattersn between our populations α/β and Worley’s populations blastema 1 and 2 (Figure 3—figure supplement 2). The integrated dataset across the two studies is now also available through our dedicated SCope session (https://scope.aertslab.org/#/WingAtlas/*/welcome). This integrative analysis shows that both bastema populations are similar to our wound : population (pro-proliferative, wg positive), in line with their identification of blastema populations as regenerative cells. We find no evidence for a similar expression pattern with our wound population β, and we suspect the difference in dissection timing between both studies to account for this effect. Indeed, Worley’s wing discs were collected 24 hours into the regeneration period, whereas our discs were collected at the end of egr induction in order to detect stress response patterns, providing us a better chance to detect the senescent-like cells β.

3. In Figure 4d, both the wound α and β populations appear to have an elevated proliferative signature compared to control pouch cells, but the authors state on line 289: "In contrast, cells from cluster β do not show clear proliferative nor regeneration markers."Can they resolve this apparent discrepancy?

The labels of Fig4d were not chosen adequately, as they should not have been named "proliferative". This was indeed confusing. We have edited the text as follows: the signature from Fig4d are now renamed more explicitly as “wound α versus control pouch signature“ and “wound β versus control pouch signature” respectively.

4. The authors should explain in the results what the left of Figure 4b is meant to illustrate. In general, this section is quite dense and difficult to follow, and would benefit from some editing.

We have edited the corresponding figure legend (line 438-440) and have reworked the corresponding result section (line 420-434).

5. Lines 303-314: the authors list several SASP markers. As this is an important part of their work, could they produce a plot showing the increase of these markers in cluster β compared with the other populations? Also, it would be good to validate at least one of these markers by staining or HCR on injured versus control wing discs.

Together with our new tumor sub-clustering analysis (cf. major issue #1), we provide a supplementary figure showing the expression levels of the upregulated SASP markers in each wound and tumor population (figure 4 – —figure supplement 2b). Concerning the experimental validation of the cluster β markers, we have performed a bulk RNA-seq experiment following the overexpression of the stress-induced isoform of Xrp1 in wing discs (line 354-363 in results, line 591-599 in methods figure 3e and figure 3 – —figure supplement 4). This experiment demonstrates the significant up-regulation of senescent markers following Xrp1 induction, which supports our eGRN inference highlighting CEBP-like TFs as potential drivers of the senescence program. Additional stainings from Jaiswal et al. are also now mentioned in the manuscript (line 545-555): they show the presence of senescent markers (e.g. β-Galactosidase) at the center of the wound in the same model. Together with the new section on Xrp1 overexpression, we have provided the complete list of SASP markers along with their ranked expression change following Xrp1 induction (figure 3 – —figure supplement 4).

Reviewer #2:The strengths of the manuscript include the generation of large-scale single-cell data sets (including single-cell transcriptomics and scRNAseq+sc-ATAC-seq), and the comparative analysis of the genes and eGRNs utilized during wound healing and tumorigenesis. The authors identify enhancers that are cell type specific through sc-ATAC-seq. Based on known transcription factor binding motifs, the authors find an enrichment for binding sites from AP-1 and STAT in enhancers utilized during both wound healing and tumorigenesis.However, there are major weaknesses, including that the manuscript covers a large number of bioinformatic comparisons without almost any experimental validation within the wound or tumor models. In the few incidences where the authors show gene expression patterns within the wounded tissues, the images are not very convincing to identify the unique cell populations, and experimental validation is completely lacking from the tumor section. At multiple points in the paper, showing experimental evidence of the unique cell populations or enhancers within the wounded and/or tumor models would significantly advance the manuscript. In addition, the manuscript needs to improve placing the findings in a larger context within the field.The comparisons between wounds and tumors that are made in this manuscript are of broad interest but many of the findings are still preliminary.1. The analysis of the wounded imaginal wing discs does show a consistent location of wound α and wound β cells. Where are these cells located? Trbl-GFP (Figure 3d) expression appears to be elevated in a ring around the wound, while upd3 (Figure 1d) is in the center of the wound. In addition, GstD9-GFP appears to be expressed within a similar number of cells as MMP1, which is not consistent with the dot plot on the right side of these images. One of the claims of the paper is that they have characterized senescent markers – however, it is not clear where these cells are located within the wounded model and no localization within the tumor models is shown.

Together with our new tumor sub-clustering analysis (cf. major issue #1), we provide a supplementary figure showing the expression levels of the upregulated SASP markers in each wound and tumor population (figure 4 – —figure supplement 2b). Concerning the experimental validation of the cluster β markers, we have performed a bulk RNA-seq experiment following the overexpression of the stress-induced isoform of Xrp1 in wing discs (line 354-363 in results, line 591-599 in methods figure 3e and figure 3 – —figure supplement 4). This experiment demonstrates the significant up-regulation of senescent markers following Xrp1 induction, which supports our eGRN inference highlighting CEBP-like TFs as potential drivers of the senescence program. Additional stainings from Jaiswal et al. are also now mentioned in the manuscript (line 545-555): they show the presence of senescent markers (e.g. β-Galactosidase) at the center of the wound in the same model. Together with the new section on Xrp1 overexpression, we have provided the complete list of SASP markers along with their ranked expression change following Xrp1 induction (figure 3 – —figure supplement 4).

2. The word "senescent", which is used throughout the manuscript to describe the wound-β cells, is intriguing but ultimately potentially misleading and preliminary. Senescent implies that a cell will never enter the cell cycle again. The EDU experiment shows a reduction of cells in S-phase within the center of the wound. However, this is not quantified, and the positions of the wound-α and wound-β cells are not shown. In addition, the use of the word senescent cellular state to describe a growing tumor model is also confusing, especially when the authors conclude that the cells are homogenous. If the cells of the tumor are homogenous and the tumor is growing through cell proliferation, then cells with an activated senescent-cellular program must also be actively dividing.

We have clarified the use of the term senescence as a description of the gene expression program rather than the terminal cell cycle arrest, where we stress the potential transience of the observed program in contrast to the terminal cell cycle arrest observed in classical senescence (line 316-319) : Nevertheless, it still remains unclear whether these wounded cells will later die or whether they will revert to the pouch disc fate after wounding (Jaiswal et al., 2022). Hence, we refer here to cellular senescence and associated secretory phenotype in terms of gene expression programs rather than evidence of a terminal cell cycle arrest.

We have also toned down the conclusion on the homogeneity level of the tumor population (line 420-434 in results, line 545-555 in discussion).

3. Along similar lines, the paper strongly emphasizes the relationship between the cellular states activated within the wound and the cellular states within a tumor model. The major hypothesis from the bioinformatics should be investigated back in the tumor model. Are the two identified pathways (JNK/AP-1 and STAT) activated uniformly within the tumor model? If this GRN activates a cellular senescent-like program, and the cells within the tumors are in fact homogenous, then what mechanisms are driving the growth of the tumor? Recent work using single-cell data has highlighted the heterogeneity within scrib imaginal-disc tumors (eg. Deng et al., 2019; Ji et al., 2019; Worley et al., 2022). What do you think explains this difference between "scrib" vs "rasV12+scrib" cells? Are "rasV12+scrib" more homogenous than scrib alone? This statement would be greatly strengthened by visualizing co-expression of the "proliferative" or "senescent-like" pathways within these tumors to provide evidence that the rasV12 scrib-/- tumors are in fact co-activating and that the cells are homogenous.

We have re-written this section, providing nuance for the seeming homogeneity in the scRNAseq data, and taking into account recent findings of tumor heterogeneity at earlier time points from Jaiswal et al., bioRxiv 2022 (line 420-434 in results, line 545-555 in discussion).

4. The manuscript does not properly acknowledge previous work on damage-responsive enhancers (e.g. the wg/Wnt6 enhancer described in Harris et al. 2016; and the enhancers identified using Bulk-ATAC-seq by Vizcaya-Molina et al. 2018 and Harris et al. 2020). How do the enhancers uncovered in this paper relate to the enhancers uncovered in this past work? For a resource paper, it would be valuable to make stronger connections to how this work connects to prior knowledge.

We agree with the reviewer that connection to existing validated damage-responsive enhancers will benefit the manuscript. We have added a supplementary figure (figure 3 – —figure supplement 1) highlighting the changes in chromatin accessibility in wound and tumor conditions at the BRV118 enhancer (which is indeed activated, showing a significant increase in accessibility, in both wound populations).

We have also contrasted our set of wound-responsive elements with the works of VizcayaMolina et al. 2018 and Harris et al. 2020 (line 222-227) and found a 24% overlap of reported enhancers within our set. We have detailed the individual enhancer overlap information as a supplementary table 3.

5. Several points of this paper are similar to a single-cell study of regenerating imaginal discs that was recently published (Worley et al., 2022), including the comparison between wounded and developing tissues, as well as the comparison of damaged and tumorous tissues. How do the wound-α and wound-β relate to the identified blastema-1 and blastema-2 cell populations?

cf. answer to reviewer 2, question #2

6. The statement that the senescent eGRN is activated and driven by a group of transcription factors needs additional experimental evidence. Driven applies that one or more of these transcription factors (Irbp18, Xrp1, etc.) would be required for the activation of these enhancers, and/or that the overexpression of one of these transcription factors would be sufficient. Experimental evidence is currently absent.

*cf.* answer to reviewer 2, question #5 :

We believe these new experiments corroborate the activity of our wound markers. Yet, the validation of their specific regulatory mechanics would require more than six months of in vivo genetic experiments that would fall outside of the scope of this study. To stress this, we agree with the reviewer that our set of markers are indeed predictive at this point, and we clarify this status in the text (line 557-558).